# Insights into the client protein release mechanism of the ATP-independent chaperone Spy

Wei He[1,5], Xinming Li[2,5], Hongjuan Xue[3], Yuanyuan Yang[1], Jun Mencius [1], Ling Bai[1], Jiayin Zhang[1], Jianhe Xu[1], Bin Wu [3,6✉], Yi Xue [2,6✉] & Shu Quan [1,4,6✉]

Molecular chaperones play a central role in regulating protein homeostasis, and their active forms often contain intrinsically disordered regions (IDRs). However, how IDRs impact chaperone action remains poorly understood. Here, we discover that the disordered N terminus of the prototype chaperone Spy facilitates client release. With NMR spectroscopy and molecular dynamics simulations, we find that the N terminus can bind transiently to the client-binding cavity of Spy primarily through electrostatic interactions mediated by the N-terminal D26 residue. This intramolecular interaction results in a dynamic competition of the N terminus with the client for binding to Spy, which promotes client discharge. Our results reveal the mechanism by which Spy releases clients independent of energy input, thus enriching the current knowledge on how ATP-independent chaperones release their clients and highlighting the importance of synergy between IDRs and structural domains in regulating protein function.

[1] State Key Laboratory of Bioreactor Engineering, East China University of Science and Technology, Shanghai Collaborative Innovation Center for Biomanufacturing (SCICB), Shanghai 200237, China. [2] School of Life Sciences, Tsinghua-Peking Joint Center for Life Sciences, Beijing Advanced Innovation Center for Structural Biology, Tsinghua University, 100084 Beijing, China. [3] National Facility for Protein Science in Shanghai, ZhangJiang Lab, Shanghai Advanced Research Institute, Chinese Academy of Sciences, Shanghai 201210, China. [4] Shanghai Frontiers Science Center of Optogenetic Techniques for Cell Metabolism, East China University of Science and Technology, Shanghai 200237, China. [5]These authors contributed equally: Wei He, Xinming Li. [6]These authors jointly supervised this work: Bin Wu, Yi Xue, Shu Quan. ✉email: wubin@sari.ac.cn; yixue@mail.tsinghua.edu.cn; shuquan@ecust.edu.cn

mbalance in protein homeostasis ("proteostasis") can lead to protein misfolding and aggregation, which is associated with a variety of human diseases such as peripheral amyloidosis, type II diabetes, cancer, cardiovascular diseases, and many neurodegenerative disorders[1–3]. To maintain proteostasis, molecular chaperone proteins, which are fundamental to protein quality control systems in cells, exert multiple functions including de novo folding of nascent polypeptides, preventing aggregation of folding intermediates, disassembling protein aggregates, and directing the degradation of off-pathway misfolded proteins[4,5].

It is estimated that 36.7% of the amino acid sequences in chaperones can be classified as "intrinsically disordered regions" (IDRs, i.e., protein regions lacking fixed tertiary structures)[6]. IDRs exist in a heterogeneous ensemble of conformations and play various roles, including protein complex assembly, post-translational modification, signal transduction, substance transportation, and molecular recognition[7–9]. The feature of molecular recognition presented by IDRs fits well with the promiscuous nature of molecular chaperones, which often need to recognize and bind to diverse kinds of proteins. Researchers have observed interactions between chaperone IDRs and client proteins and have noted the functional importance of IDRs in chaperone actions[10]. For example, the disordered C terminus of DnaK provides a binding motif with weak affinity for the clients, maintaining the ability of DnaK to support bacterial growth under thermal stress[11]. The disordered C terminus of GroEL actively disrupts kinetically trapped, locally misfolded regions in the client proteins to facilitate their refolding[12]. Several "conditionally disordered" chaperones, such as Hsp26, Hsp33, and HdeA, rely on order-to-disorder conformational changes to bind and protect various clients from stress conditions[13].

Despite increased research on the function of IDRs in chaperones, detailed kinetic and structural characterization of IDRs in chaperones is still lacking. A major challenge is that chaperone clients are prone to aggregation, which limits kinetic and structural analysis. In addition, the high flexibility of IDRs hinders the study of IDRs by traditional biochemical and biophysical methods. Nuclear magnetic resonance (NMR), on the other hand, has the unique advantage of providing detailed residue-by-residue information of protein disorder, making it highly suitable for investigating dynamic IDRs in chaperones[14]. However, high-molecular-weight proteins are a challenge to resolve by NMR and thus, it has been difficult to structurally elucidate the roles of IDRs in canonical chaperones, such as Hsp70 and GroEL.

The ATP-independent chaperone Spy harbors long intrinsically disordered N and C termini, and their influence on Spy's chaperone action remains enigmatic. The dimeric protein Spy has a molecular weight of 32 kDa and is a good model for structurally understanding the mechanism of IDRs in chaperone action by NMR spectroscopy techniques. Spy was initially identified in a genetic screen of chaperones to stabilize a protein called Im7 in the bacterial periplasm, and overexpression of Spy increased the steady-state level of Im7 up to 700-fold[15]. Im7 is a model protein for protein-folding studies, and mutants of Im7 trapped in fully or partially unfolded states have been constructed and validated as highly soluble[16–18]. Therefore, we used this model chaperone-client pair as well as a few other well-characterized Spy client proteins[19–22] to address the fundamental questions of whether and how dynamic IDRs affect chaperone action from a kinetic and structural perspective.

Here, we show that the disordered N terminus of Spy facilitates client protein release. Data from mutagenesis studies, kinetic analyses, NMR spectroscopy, and molecular dynamics (MD) simulations together support that the N terminus can bind transiently to the concave, client-binding surface of Spy, primarily through electrostatic interactions between the N-terminal D26

residue and five positively charged residues on the concave surface. In addition, the binding sites of the N terminus on the concave surface of Spy partially overlap with the binding sites of client proteins. As a result, the N terminus can facilitate client protein release by competitively binding to Spy's concave surface. Furthermore, the client release-promoting effect of the N terminus depends on its conformational restriction by adjacent regions in the full-length protein, since the peptide with the same sequence as the N terminus but free in solution cannot achieve this effect. Our work expands the existing working mechanism of Spy and reveals tantalizing new features of intrinsic disorder in chaperone action.

## Results

**The N terminus of Spy contributes to client release.** Both the N and C termini of Spy (residues 1–28 and 125–138, respectively) are predicted to be highly disordered and are not visible in the crystal structure of Spy (Fig. 1a and Supplementary Fig. 1a). To investigate the effects of disorder on Spy's chaperone activity, we deleted one or both termini of Spy and obtained the variants Spy$_{29-138}$, Spy$_{1-124}$, and Spy$_{29-124}$. We also started from Spy$_{21-130}$, a previously characterized truncation mutant that retains the ability to stabilize Im7 in vivo[19], and systematically shortened its termini, three residues at a time, to obtain a series of terminally truncated Spy variants (Fig. 1b).

We observed little change in Spy's overall secondary structure after termini removal or truncation (Supplementary Fig. 1b). We then determined the ability of these variants to prevent the aggregation of dithiothreitol-reduced α-lactalbumin (α-LA). We found that deletion of the entire C terminus (Spy$_{1-124}$) did not affect the anti-aggregation activity of Spy, but the absence of the entire N terminus (Spy$_{29-138}$, Spy$_{29-124}$) increased the activity of Spy by at least 1.7-fold (Fig. 1c). Additionally, removal of up to 23 residues from the N terminus resulted in moderately decreased activity, but further shortening by 26 residues resulted in an abrupt increase in Spy's activity. In contrast, C-terminal deletion had a lower effect on Spy's chaperone activity and there was no apparent correlation between the change in activity and the extent of C-terminal truncation (Fig. 1c). These results thus suggest that the N-terminal residues, rather than C-terminal residues, have a decisive influence on the chaperone activity of Spy.

We previously isolated activity-enhancing "Super Spy" variants through a genetic selection that coupled Spy's ability to stabilize an unstable mutant of Im7 with antibiotic resistance of E. coli[20]. These "Super Spy" variants all sacrificed their own stability for flexibility, thereby enhancing chaperone activity[20]. To investigate whether a similar strategy is adopted by the terminally truncated Spy variants constructed in this study, we measured their unfolding Gibbs free energy ($\Delta G_{NU}$) but found no obvious correlation between the stability and activity of these variants (Supplementary Fig. 1c), suggesting that changes in overall flexibility could not explain the activity differences of these variants.

To understand how terminal truncation affects Spy activity, we determined the association and dissociation kinetics of the Spy variants with the partially folded client, Im7 H40W L53A I54A (Im7$_{AAW}$)[22], by biolayer interferometry. We found significant differences in the dissociation constant ($K_d$) and the dissociation rate constant ($k_{off}$) between the terminal-truncated variants. Variants that showed higher anti-aggregation activities toward α-LA with the removal of 26 or 28 residues of the N terminus also displayed tighter binding to Im7$_{AAW}$ and slower release of Im7$_{AAW}$ (Fig. 1d, e and Supplementary Fig. 1d–i). In contrast, the association rate constants ($k_{on}$) of most variants were comparable to that of the Spy wild type (Fig. 1f). These results suggest that the

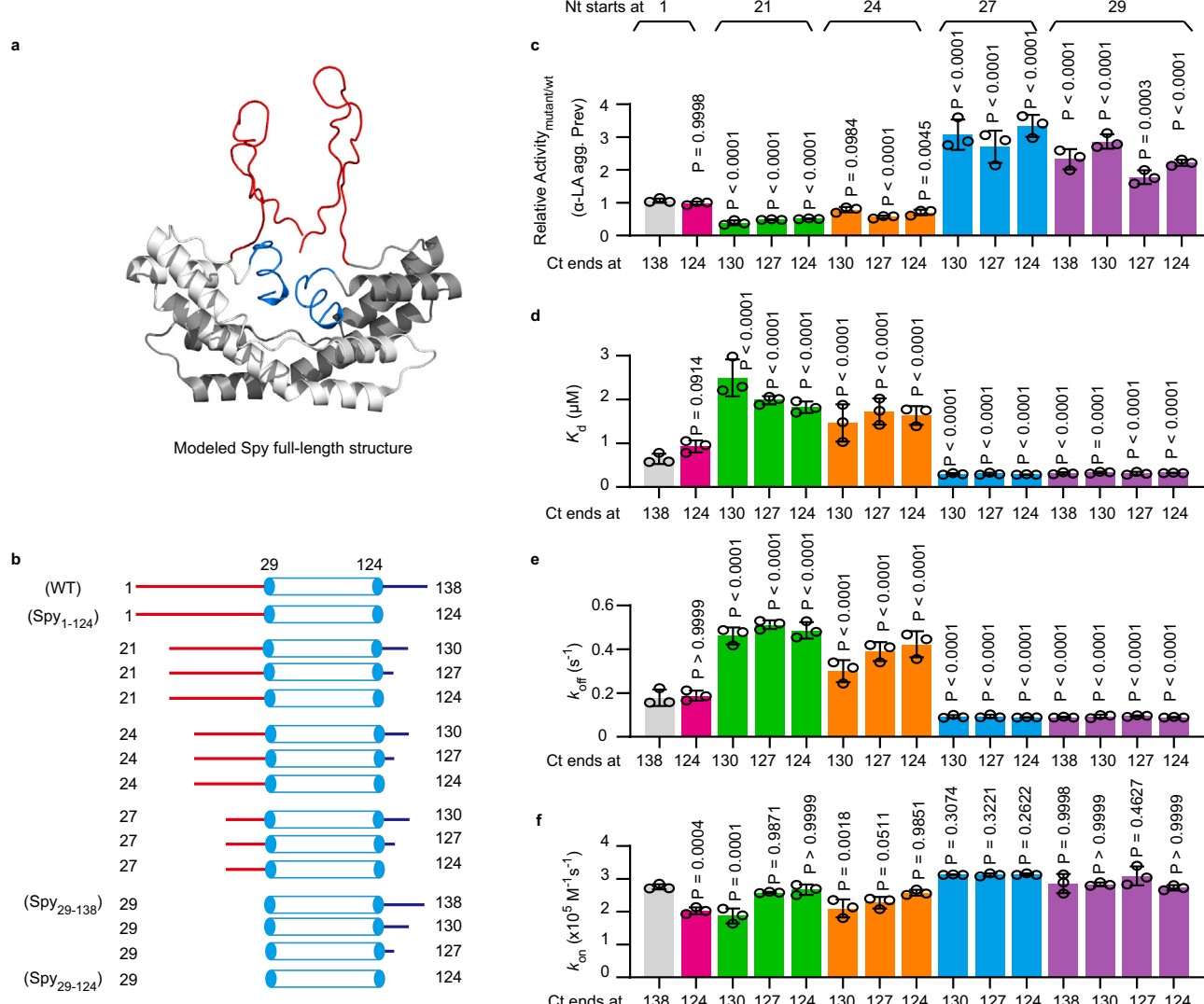

**Fig. 1 The disordered N terminus of Spy is involved in client protein release. a** Structure model of full-length Spy with the N- and C-termini colored in red and blue, respectively. The disordered termini were modeled using the CABS-dock software and docked to the crystal structure of Spy (PDB: 3O39), in which only residues 29–124 are visible[15]. For clarity, only one possible conformation of the termini is shown. **b** Schematic illustration of termini-truncated Spy variants used for the chaperone activity assay and kinetics assessment, where the numbers listed as suffixes describe the amino acids remaining in the variants, using this nomenclature wild type Spy is Spy$_{1-138}$. The structured regions of Spy$_{29-124}$ are represented by blue cylinders, and the N- and C-termini are shown as red and blue lines, respectively. **c** Chaperone activities of termini-truncated Spy variants in inhibiting the aggregation of DTT-reduced α-LA relative to the activity of Spy wild type. Abbreviation α-LA agg. Prev means α-LA aggregation Prevention. Comparison of the dissociation constants $K_d$ (**d**), dissociation rate constants $k_{off}$ (**e**), and association rate constants $k_{on}$ (**f**) between termini-truncated Spy variants using the model client protein Im7$_{AAW}$ as the chaperone substrate. For **c–f**, individual data points (circles) and mean ± SD ($n = 3$ independent experiments) are shown. Statistical analysis was performed with Graphpad Prism 9.1.0 using one-way ANOVA with Tukey's multiple comparisons test. Source data are provided as a Source Data file.

long and disordered N terminus of Spy may not be directly involved in the initial binding process of the client, but may contribute to the client release in a length- and composition-dependent manner.

To test whether the facilitation of client release by Spy's N-terminus also applies to other clients and, more importantly, to clients with different folding states, we measured the binding and release kinetics of Spy wild type and Spy$_{29-138}$ toward two unfolded clients, namely carboxymethylated α-LA and Im7 L18A L19A L37A (Im7$_{A3}$), and one folded client, Im7 wild type. We found that for all clients, Spy$_{29-138}$ exhibited slower client-release rates, slightly decreased association rates, and overall tighter affinity compared to those of the Spy wild type (Supplementary Fig. 2). These results thus highlight the generality of Spy's N terminus in facilitating client release.

**D26 is the key residue in facilitating client release**. The removal of 20 or 23 residues from Spy's N terminus gave rise to a ~2-fold increase in substrate release rates, while removal of 26 or 28 residues resulted in a sudden decrease in client release rates (Fig. 1e), suggesting that the three residues (H24, Q25, and D26) may play a critical role in facilitating client release. We previously isolated an activity-enhancing point mutation of Spy that substituted Q25 with an arginine residue[20]. Not coincidentally, the Q25R mutant also showed a 2.3-fold decrease in $k_{off}$ rate and a 1.7-fold decrease in $k_{on}$ rate, which resulted in a 1.3-fold increase in binding affinity[20]. Therefore, the presence of a positively charged residue at position 25 appears to counteract the effect of H24, Q25, and D26 in facilitating client release. The release-promoting effect of residues H24, Q25, and D26 was even more pronounced after the removal of N-terminal residues 1–23

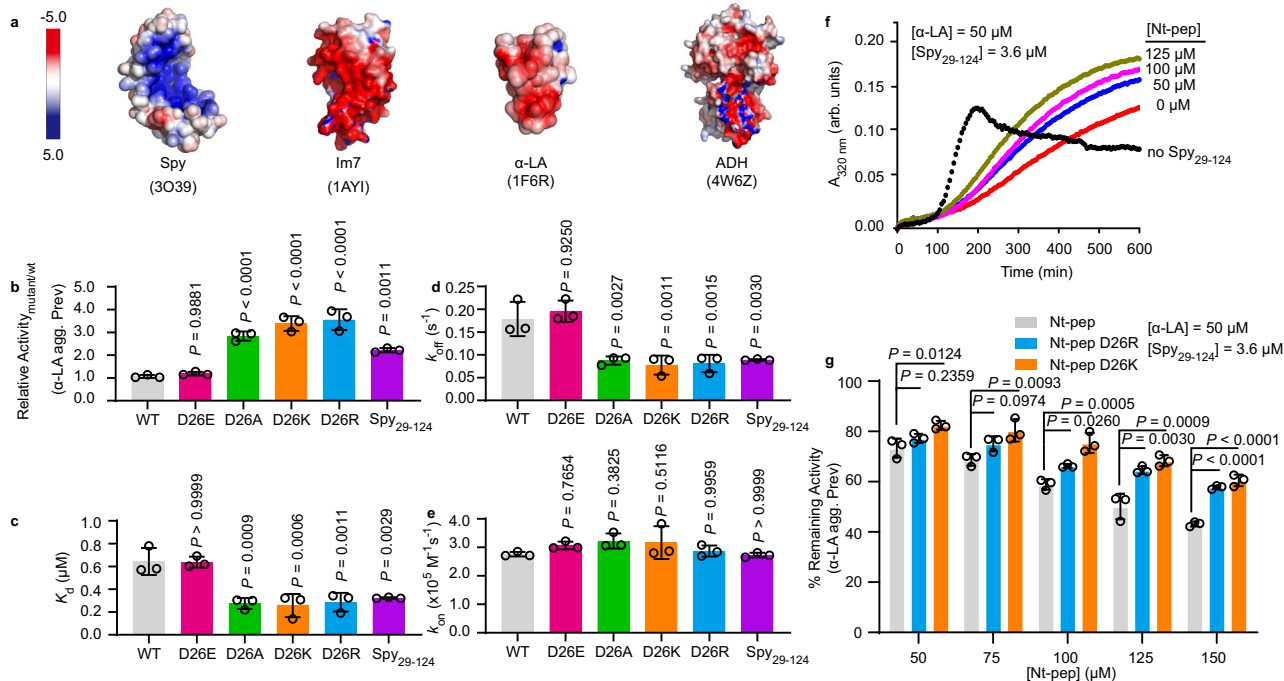

**Fig. 2 The D26 residue located on Spy's disordered N terminus facilitates client release. a** Surface charge distribution of Spy and model client proteins. The concave, client-binding surface of Spy (PDB: 3O39) is predominantly positively charged (blue), while the surfaces of client proteins Im7 (1AYI), α-LA (1F6R), and ADH (4W6Z) are generally negatively charged (red). The electrostatic surface potential was calculated in PyMOL with the Adaptive Poisson-Boltzmann Solver tools 2.1 plugin. A color scale for the charge distribution ranging from –5 to 5 was applied. **b** Chaperone activities of different Spy variants in preventing the aggregation of DTT-reduced α-LA relative to the activity of Spy wild type. Kinetic parameters characterizing the interactions between Spy variants and $Im7_{AAW}$, including the dissociation constants $K_d$ (**c**), dissociation rate constants $k_{off}$ (**d**), and association rate constants $k_{on}$ (**e**). **f** The anti-aggregation activity of $Spy_{29-124}$ in the presence and absence of various concentrations of Nt-pep. Representative curves from three independent measurements are shown. In the absence of $Spy_{29-124}$, massive aggregation of α-LA resulted in the decline of the light scattering signal at 320 nm after 200 min (black curve). **g** Quantification for remaining anti-aggregation activities of $Spy_{29-124}$ in the presence of various concentrations of Nt-pep, Nt-pep D26K, and Nt-pep D26R. For **b**–**e**, **g** individual data points (circles) and mean ± SD (n = 3 independent experiments) are shown. Statistical analysis was performed with Graphpad Prism 9.1.0 using one-way ANOVA with Tukey's multiple comparisons test. α-LA agg. Prev means α-LA aggregation Prevention. Source data are provided as a Source Data file.

(Fig. 1e), suggesting that residues 1–23 are also able to counteract the effects of H24, Q25, and D26. Supporting this idea, we found five positively charged residues (K12, H16, H17, K18, and K20) located in the N terminal region prior to residues 24–26.

The interaction between Spy and its client proteins is mediated by both hydrophobic and electrostatic forces[19,22]. Spy is a cradle-shaped molecule with an overall positively charged, concave surface for binding to clients, whereas many clients, such as Im7, α-LA, and ADH, are overall negatively charged (Fig. 2a). Since D26 is the only negatively charged residue among the 24th–26th residues, we hypothesize that D26 is the residue that contributes the most to client release. Possible speculation for the mechanism by which the N terminus facilitates client release is that the N terminus containing the negatively charged D26 competes with the client for binding to the concave surface of Spy, which may destabilize the Spy-client complex and facilitate the detachment of the client from Spy's surface.

To test this hypothesis, we first replaced D26 with a negatively charged residue, glutamate (D26E), a neutral residue, alanine (D26A), or positively charged residues, lysine or arginine (D26K or D26R), and then examined the chaperone activity and substrate binding kinetics of the four variants. We found that D26E showed nearly the same activity and binding kinetics as Spy wild type, whereas variants D26A, D26K, and D26R exhibited enhanced activities compared with Spy wild type (Fig. 2b). More importantly, D26A, D26K, and D26R decreased $k_{off}$ and $K_d$ to levels comparable to the termini-free $Spy_{29-124}$ variant without

significant changes in the $k_{on}$ rates (Fig. 2c–e). These results suggest that the presence of a negative charge at the 26th position facilitates the release of Spy clients but does not interfere with the initial client-binding process. Electrostatic interactions depend on the distance between two objects. Thus, we reckoned that the release-promoting effect of D26 might be attenuated by inserting a linker between residue D26 and M27, which increases the distance between D26 and Spy's concave surface. Indeed, insertion of the $(GGGS)_2$ linker reduced the $k_{off}$ rate and led to an increase in Spy's chaperone activity (Supplementary Fig. 3a–c).

To obtain evidence that the D26-containing N terminus of Spy can compete with clients for binding to Spy's concave surface, we monitored the aggregation kinetics of α-LA in the presence of $Spy_{29-124}$ and the peptide corresponding to Spy residues 1–28 (hereinafter referred to as Nt-pep). Nt-pep itself had no effect on the aggregation kinetics of α-LA (Supplementary Fig. 3d). In contrast, the addition of Nt-pep in the presence of $Spy_{29-124}$ reduced the ability of Spy to prevent α-LA aggregation in a dose-dependent manner (Fig. 2f), indicating that Nt-pep had an inhibitory effect on Spy's activity. The same effect was not observed when increasing concentrations of the C-terminal peptide (corresponding to Spy residues 125–138, Ct-pep) was added to Spy (Supplementary Fig. 3e). Additionally, Nt-pep containing the D26K or D26R mutation showed a weaker inhibitory effect (Fig. 2g), underscoring the role of D26 in mediating the interaction between Nt-pep and Spy. Based on these results, we suggest that D26 may have direct electrostatic

contact with the concave surface of Spy, leading to competition between the N terminus and negatively charged clients, thus affecting the release of clients from Spy.

**The N terminus approaches Spy's concave surface**. To obtain structural insight into the above hypothesis, we performed MD simulations. We found that the N terminus of Spy readily swings into the positively charged, client-binding cavity of Spy, regardless of the starting conformation, and in the presence or absence of C terminus in the MD simulations. In contrast, substituting residue D26 with a positively charged arginine resulted in limited access of the N terminus to the concave surface (Supplementary Movies 1 and 2, Supplementary Fig. 4, and Supplementary Data 1).

To experimentally test our MD simulation results and to elucidate the detailed intramolecular contacts between the N terminus and Spy's structured region, we performed intramolecular paramagnetic relaxation enhancement (PRE) experiments on $^{15}$N-labeled Spy$_{1-124}$, where the paramagnetic spin label methanethiosulfonate (MTSL) was attached to a cysteine inserted adjacent to D26 or one residue away, i.e., immediately before Q25, D26, M27, or M28 (Fig. 3a and Supplementary Fig. 5a). The PRE phenomenon arises from magnetic dipolar interactions between unpaired electrons at the paramagnetic center and nearby nuclei (typically, backbone amide protons), which can provide long-range distance information up to 35 Å[23].

Following the backbone resonance assignments of Spy$_{1-124}$ (Fig. 3b), we found that all the MTSL spin labels around residue D26 had strong effects on many residues of Spy$_{1-124}$, with the spin label attached to the cysteine inserted between residues Q25 and D26 (hereinafter referred to as CI) having the strongest effect, resulting in the highest PRE intensity ratios (Fig. 3c and Supplementary Fig. 5b–d). In contrast, attaching the MTSL label to the cysteine substitution of M15 or G19, which is more than six residues away from D26, led to small PRE intensity ratios (Supplementary Fig. 5e, f). These results indicate that residues near D26 make extensive transient contacts with the structured region of Spy.

Furthermore, based on the PRE effect at CI, we noticed that all of the positively charged residues with side chains located on the concave surface of Spy (blue bars in Fig. 3c) have a PRE intensity ratio higher than 2, except for residue R43, whose PRE intensity ratio could not be calculated due to peak overlap. In contrast, only two of the eight positively charged residues with side chains pointing to the convex surface (black bars in Fig. 3c) have a comparable PRE signal. Therefore, these results support that the N terminus is predominantly approaching the concave surface of Spy. We then mutated residues Q25 and D26 on Spy$_{1-124 CI}$ to arginine and lysine, respectively (Fig. 3a), to assess the influence of the negative charge at position 26 on the accessibility of the N terminus to the concave surface of Spy. The PRE effects of MTSL attached at position CI of the mutant protein were dramatically reduced (Fig. 3c). This result well supports our speculation based on biochemical data (Fig. 2) that the negative charge of D26 is the key factor mediating intramolecular contacts between Spy's N terminus and its concave surface.

To investigate whether the other two N-terminal negatively charged residues D2 and D10 also form electrostatic contacts with the concave surface of Spy, we introduced the MTSL spin label on a cysteine substitution at T5 of $^{15}$N-labeled Spy$_{1-124}$ (Fig. 3a) and determined the intramolecular PRE effect. We found that the spin label attached to T5C affected many residues on Spy$_{1-124}$, with positively charged residues on Spy's concave surface being more affected than those on the convex side (Fig. 3c). Similarly, substituting D2 and D10 with positively charged lysine residues

decreased the PRE intensity ratios (Fig. 3c). However, we also note that the PRE effect of the spin label attached to T5C was much weaker than the signal obtained when the spin label was placed at position CI. In addition, the D2K D10K mutation had a negligible influence on the PRE effect of the spin label attached to CI, but Q25R D26K dramatically decreased the PRE intensity ratios of the spin label attached to T5C (Fig. 3c). Together, these results suggest that although D2 and D10 can also approach the concave surface of Spy, their influence is much smaller than that of D26, the key residue that dictates the long-range electrostatic interactions between the N terminus and the structured region of Spy.

**D26 interacts with key residues on Spy's concave surface**. We next set out to identify positively charged residues that might interact electrostatically with D26, focusing on all the arginine or lysine residues located on the central cavity of Spy (Fig. 4a). We reasoned that if we mutated the residue interacting with D26 to aspartate, we could reverse the electrostatic attraction between D26 and this position to repulsion, thus reducing the PRE effect of the spin label of CI on the cavity region. Therefore, we substituted these residues on $^{15}$N-labeled Spy$_{1-124 CI}$ and examined the PRE intensity ratios of the resulting five single point mutants (R43D, R50D, H65D, R89D, and H96D) and three double mutants (K54D R55D, R61D R62D, and K121D R122D) (Fig. 4b). In addition, the K75D was chosen as a negative control because the side chain of this residue points to the convex side of Spy, and its peak intensity is nearly not affected by the MTSL spin label attached to CI (Fig. 3c).

For eight of the nine variants, their TROSY spectra remained nearly unchanged, so most of their backbone resonances could be assigned according to the assignment information of Spy$_{1-124}$ (Fig. 3b). However, the backbone resonance of Spy$_{1-124 CI H65D}$ could not be confidently assigned due to extensive chemical shift perturbations (CSPs). More importantly, these extensive CSPs suggest a non-trivial change in the overall structure of Spy. Therefore, we excluded H65D from further analysis. For accuracy, we only calculated the PRE intensity ratios of the remaining eight variants for residues showing non-overlapping resonance peaks.

We plotted the correlation of the PRE intensity ratios between the eight variants and Spy$_{1-124 CI}$, and calculated the root-mean-square deviation (RMSD) values (Fig. 4b). Based on the correlation of PRE signals of Spy residues, we classified these variants into four categories. The data points for Spy$_{1-124 CI R89D}$ tends to be distributed below the diagonal compared with other variants, indicating overall increased distances between the MTSL label and its cavity region. The data points of Spy$_{1-124 CI K54D R55D}$ and Spy$_{1-124 CI R61D R62D}$ show the largest RMSD values (1.98 and 2.23, respectively) and data points deviate from the diagonal in both directions, indicating a considerable variation in the distance distribution between the MTSL label and their structured regions. The RMSD values for Spy$_{1-124 CI R43D}$, Spy$_{1-124 CI H96D}$, and Spy$_{1-124 CI K121D R122D}$ are moderately high (1.23–1.74), indicating some variation in the distance distribution. The last category includes Spy$_{1-124 CI R50D}$ and Spy$_{1-124 CI K75D}$, which show the lowest RMSD values (0.89 and 0.78, respectively), suggesting similarity to Spy$_{1-124 CI}$. Based on these results, we speculated that R89, K54, R55, R61, and R62 are the most promising residues to interact with D26, while R43, H96, K121, and R122 also possess the potential to be D26-interacting sites. Because we saw considerable variation in the distance distribution of the MTSL label, it is possible that D26 interacts with several positively charged residues on the concave surface. The disruption of the interaction between D26 and one such residue may be partially

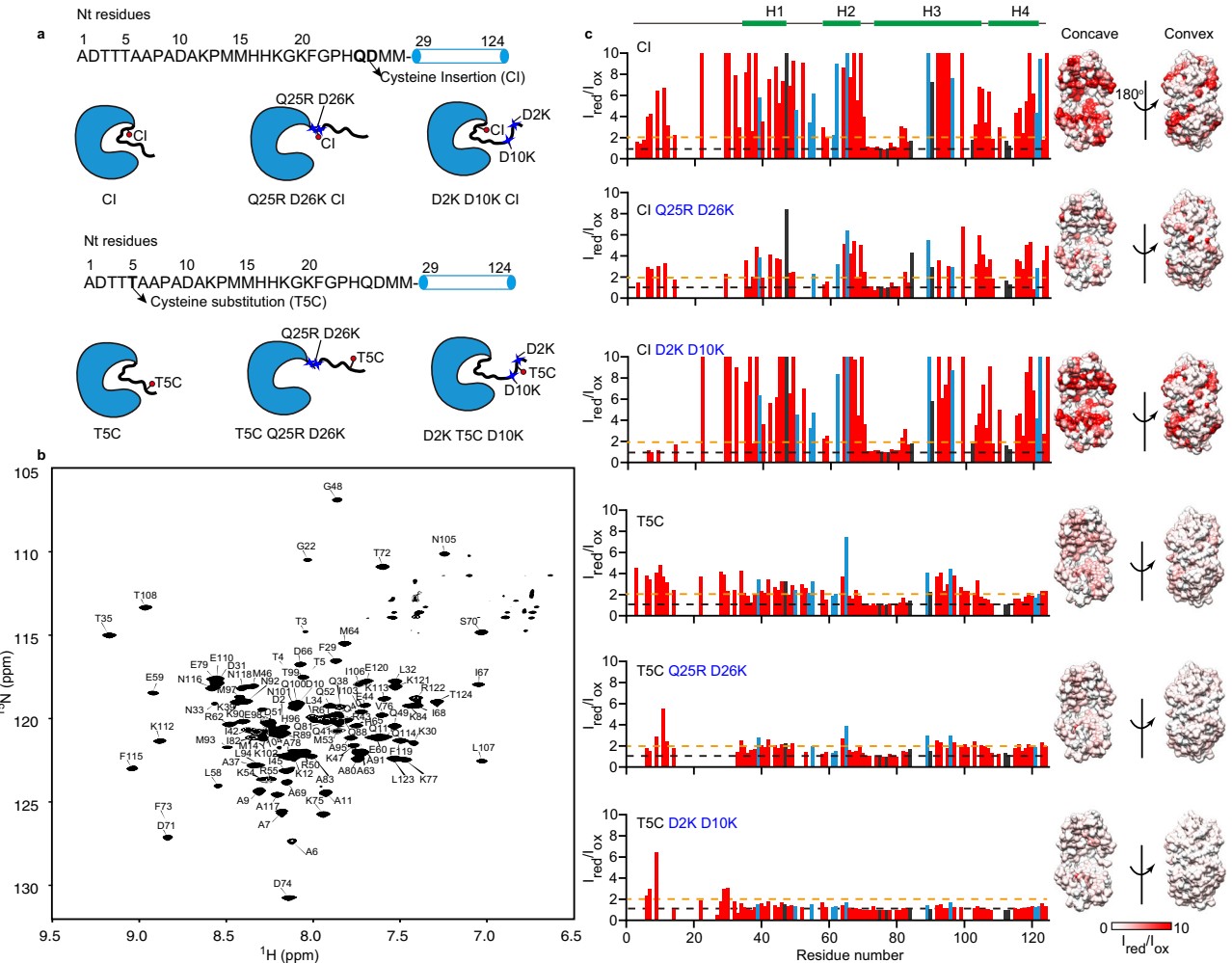

**Fig. 3 Electrostatic interactions mediate the swing of the N terminus of Spy toward its positively charged concave surface. a** Illustrations of Spy variants used in intramolecular PRE experiments. The positions for cysteine insertion or substitution are indicated by arrows. The diagrams are schematic illustration for the different positions of the N terminus relative to the structured region of Spy due to various mutations on Spy's N terminus. **b** The two-dimensional (2D) [$^{15}$N,$^1$H]- transverse relaxation-optimized spectroscopy (TROSY) spectrum of 400 μM [$^2$H, $^{13}$C, $^{15}$N]-labeled Spy$_{1-124}$ in a solution of 40 mM HEPES, 150 mM NaCl with 0.03% NaN$_3$ (w/v) and 10% D$_2$O, pH 7.5. Sequence-specific resonance assignments of the backbone amide groups are labeled. **c** Intramolecular PRE effect of a spin label on $^{15}$N-labeled Spy$_{1-124}$ variants with a cysteine inserted after residue Q25 (CI) or substituted for T5 (T5C) along with combinatorial substitutions of Q25R D26K or D2K D10K. A paramagnetic spin label (MTSL) was attached to CI or T5C. The PRE effects were presented by the ratios of peak intensities before and after the reduction of the spin label ($I_{red}/I_{ox}$). PRE intensity ratios of 1 and 2 are indicated by black and orange dashed lines, respectively. Positively charged residues with side chains pointing to the concave or convex surfaces of Spy are marked in light blue and black, respectively. Surface presentations of Spy$_{29-124}$ (PDB: 3O39), colored according to the PRE intensity ratios are displayed to the right. The white-to-red color scale indicates PRE intensity ratios ($I_{red}/I_{ox}$) ranging from 0 to 10. Source data are provided as a Source Data file.

compensated by enhancing the interaction between D26 and another such residue, which is in accordance with the inherent flexibility of the N terminus.

To further examine the roles of the above residues in mediating the interaction between the N terminus and the structured region of Spy, we substituted them with aspartate in Spy wild type and Spy$_{29-124}$, respectively, and determined the binding kinetics of these variants toward Im7$_{AAW}$. In the absence of the N terminus, all of these aspartate variants decreased the $k_{on}$ rates and increased the $k_{off}$ rates, resulting in 2.9-8.6 fold increases in the $K_d$ (Supplementary Fig. 6). Thus, these positively charged residues are likely to involve in client binding and retention. In the presence of the N terminus, we still observed increases in the $k_{off}$ rates for Spy$_{R43D}$, Spy$_{R50D}$, Spy$_{H96D}$, and Spy$_{K121D\ R122D}$ compared to Spy wild type, which could be explained by the reduced retention of clients by these mutants (Fig. 4c). However, we observed decreased $k_{off}$ rates for Spy$_{K54D\ R55D}$, Spy$_{R61D\ R62D}$,

and Spy$_{R89D}$ (Fig. 4c), suggesting that these mutations also impede the ability of the N terminus to facilitate client release, which overweighs the reduction in client retention by these mutants.

We then mutated the D26 residue to lysine in the Spy$_{K54D\ R55D}$, Spy$_{R61D\ R62D}$, and Spy$_{R89D}$ variants to test whether re-establishing electrostatic interactions involving the 26$^{th}$ residue could restore the client release-promoting effect of the N terminus in these mutants. Indeed, we found that the $k_{off}$ rate of the double mutant Spy$_{R89D\ D26K}$ was similar to that of Spy wild type, whereas the release rates of the triple mutants Spy$_{D26K\ K54D\ R55D}$ and Spy$_{D26K\ R61D\ R62D}$ were even higher than the rate of Spy wild type (Fig. 4c). For Spy$_{R43D}$, Spy$_{R50D}$, Spy$_{H96D}$, Spy$_{K121D\ R122D}$, and the control protein Spy$_{K75D}$, the additional D26K mutation decreased the $k_{off}$ rates, similar to the change in Spy$_{D26K}$ (Figs. 2d and 4c), indicating that these residues have no significant impact on the client release process mediated by the N terminus.

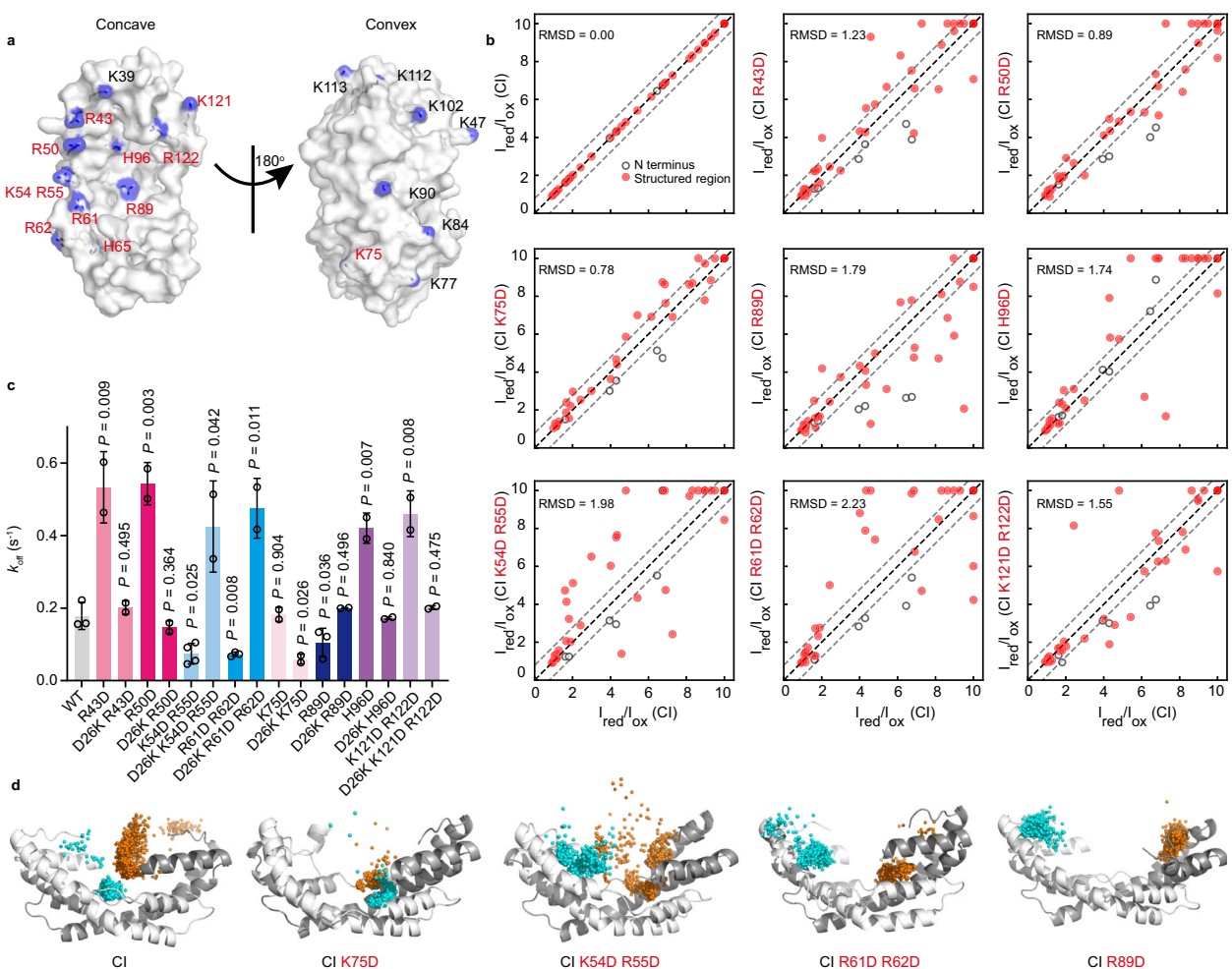

**Fig. 4 The N-terminal residue D26 interacts with residues R89, K54, R55, R61, and R62 on the concave surface of Spy. a** Surface representations of Spy$_{29-124}$ (PDB: 3O39) showing the side chains of all the positively charged residues in the stick model. The side chain nitrogen atoms of these residues are colored blue. Residues selected for further testing are labeled in red. **b** Correlations between the intramolecular PRE intensity ratios of different Spy$_{1-124\ CI}$ variants and the PRE signals of Spy$_{1-124\ CI}$. The RMSD values of the PRE intensity ratios are presented for each comparison group. Gray dashed lines represent the deviation from the diagonal (black dashed line) for a value of 0.78, which is the RMSD value between Spy$_{1-124\ CI}$ and the control protein Spy$_{1-124\ CI\ K75D}$. For the plot of Spy$_{1-124\ CI}$ and Spy$_{1-124\ CI\ K75D}$, most of the data points are distributed between the two gray lines. Thus, these lines help to make a visual evaluation for the distribution of data points. **c** Comparisons of dissociation rate constants $k_{off}$ of Spy variants to the $k_{off}$ of Spy wild type (mean ± SD, $n = 3$ independent experiments for WT, R61D R62D, and R89D, $n = 4$ independent experiments for K54D R55D, $n = 2$ independent experiments for the other variants, individual data points are shown; unpaired two-tailed Student's $t$ test). Statistical analysis was performed with Graphpad Prism 9.1.0. **d** Position of the MTSL spin label on the concave surface of Spy during MD simulations of Spy$_{CI}$ and its variants. The two monomers of Spy are shown in white and gray, respectively. A total of 1000 frames are uniformly resampled in each 1-μs trajectory, and the nitroxide oxygen (O1) atoms from each monomer are represented by cyan and orange spheres, respectively. Source data are provided as a Source Data file.

Furthermore, we performed MD simulations for several representative aspartate variants to visualize the distribution of the MTSL label in the cavity region and provide a possible picture for dynamics of the N terminus in the Spy cavity. Interestingly, we found that the MTSL label on Spy$_{CI}$ and the control protein Spy$_{CI\ K75D}$ tended to reside in the central cavity of Spy. In contrast, aspartate substitutions of K54/R55, R61/R62, or R89 on Spy$_{CI}$ resulted in the relocation of MTSL label toward the edge of Spy's cavity (Fig. 4d and Supplementary Data 1). This effect was particularly pronounced for the Spy$_{CI\ R89D}$ variant, consistent with the PRE result (Fig. 4b). To evaluate the robustness of these MD results and to confirm that the C-terminal tail does not interfere with the interaction between the N terminus and the Spy cavity, we also performed MD simulations on Spy$_{1-124\ CI}$ and Spy$_{1-124\ CI\ R89D}$ by using different starting conformations for the N terminus, all with the C terminus removed (see "Methods"). We found that all the four R89D trajectories showed clear

tendency of pushing the MTSL spin label out of the bottom of the cavity (Supplementary Fig. 7 and Supplementary Data 1). Thus, based on these experimental and simulation results, we conclude that R89, K54, R55, R61, and R62 are the primary sites on the concave surface that mediate the interactions with D26 on the N terminus.

**The N terminus acts in a competitive mode**. We have shown that the Spy residues involved in electrostatic interactions with the N-terminal residue D26 are also involved in client binding of Im7$_{AAW}$ (Fig. 4 and Supplementary Fig. 6). This raises the possibility that the N terminus of Spy and the client share overlapping binding regions. To map and compare the two binding interfaces, we collected different NMR spectra and calculated the CSPs of Spy residues.

To identify the binding sites of client proteins on Spy, we assigned the 2D [$^{15}$N, $^{1}$H]-TROSY spectra of Spy$_{29-124}$ in its apo

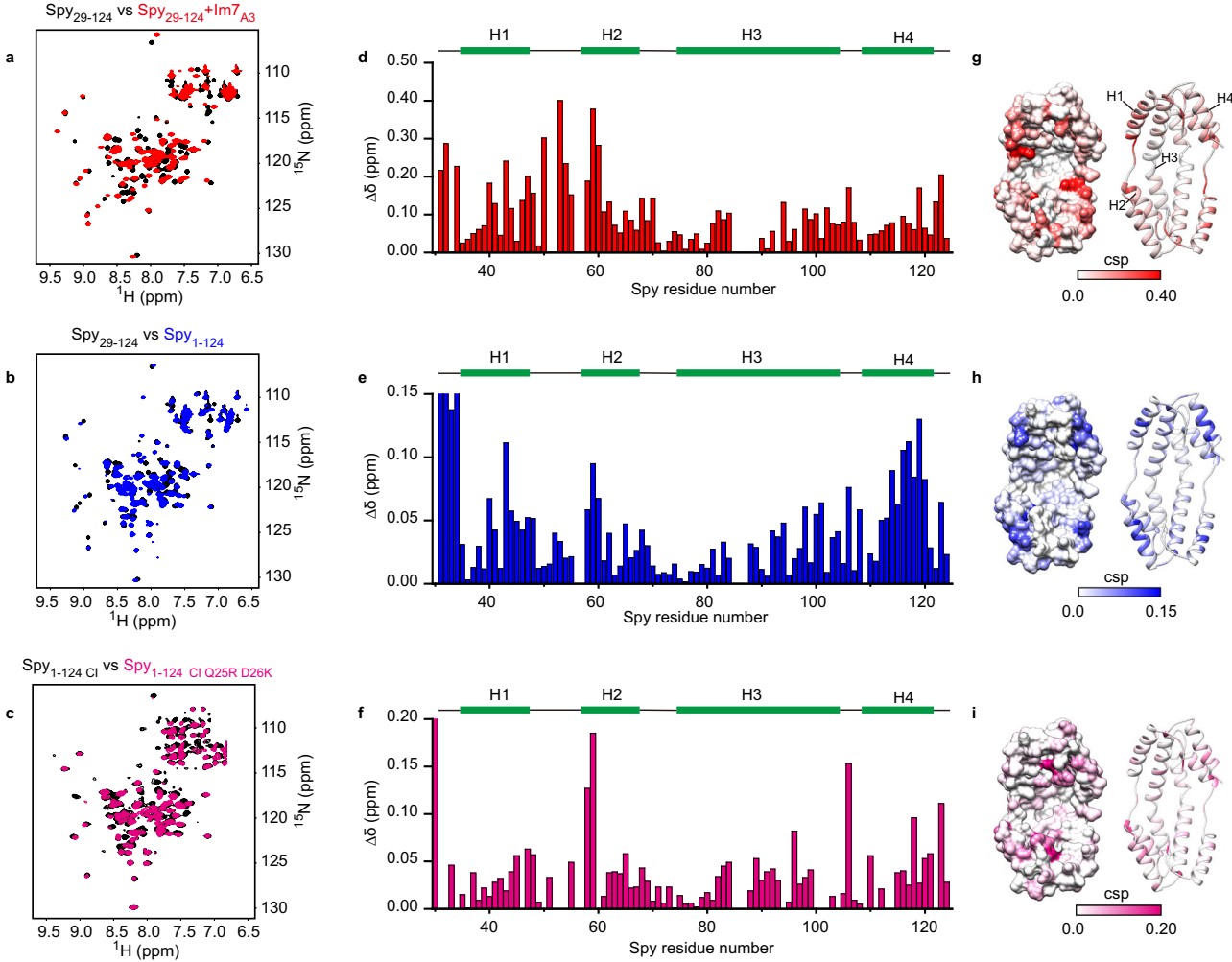

**Fig. 5 The N terminus of Spy adopts a similar binding pattern as the Im7 client on Spy's surface. a** Two-dimensional [$^{15}$N,$^1$H]-TROSY spectra of 400 μM [$^2$H, $^{13}$C, $^{15}$N]-labeled Spy$_{29-124}$ in the absence (black) and presence (red) of 400 μM unlabeled Im7$_{A3}$. **b** An overlay of the 2D [$^{15}$N,$^1$H]-TROSY spectra of 400 μM [$^2$H, $^{13}$C, $^{15}$N]-labeled Spy$_{29-124}$ (black) and 400 μM [$^2$H, $^{13}$C, $^{15}$N]-labeled Spy$_{1-124}$ (blue). **c** An overlay of the 2D [$^{15}$N,$^1$H]-TROSY spectra of 400 μM [$^{15}$N]-labeled Spy$_{1-124\ CI}$ (black) and 400 μM [$^{15}$N]-labeled Spy$_{1-124\ CI\ Q25R\ D26K}$ (pink). Chemical shift perturbations (CSPs) of amide moieties of Spy$_{29-124}$ upon binding with Im7$_{A3}$ (**d**) or upon fusion of the N terminus (**e**) were calculated and plotted against the residue number of Spy$_{29-124}$. **f** CSPs of amide moieties of Spy$_{1-124\ CI}$ upon substitution of residues Q25 and D26 by arginine and lysine, respectively. For **d**–**f**, the secondary structures of Spy$_{29-124}$ or Spy$_{1-124\ CI}$ are indicated on top, with green bars representing α-Helices. **g**–**i** Structural presentations of Spy$_{29-124}$ (PDB: 3O39), colored according to the CSP values corresponding to **d**–**f** using a two-color scale. Color intensities correlate with CSP values, with darker colors indicating larger values. Source data are provided as a Source Data file.

form and in complex with Im7$_{A3}$ (Supplementary Fig. 8a, b). We found that complex formation did not alter the secondary structure of Spy$_{29-124}$ (Supplementary Fig. 8c). The binding site of Im7$_{A3}$ on Spy$_{29-124}$ was mapped by CSPs, and the Spy residues showing the strongest CSPs were considered to be part of the binding interface (Fig. 5a, d, g).

To determine the binding site of Spy's N terminus, we used two approaches. In the first approach, we obtained the CSPs of Spy residues by comparing the 2D [$^{15}$N, $^1$H]-TROSY spectra of Spy$_{29-124}$ and Spy$_{1-124}$ (Fig. 5b, e). The presence of the N terminus affects most of Spy's residues in the structured region except for the residues located at the bottom of the concave surface and those pointing toward the convex surface (Fig. 5h). The CSPs are not due to changes in the secondary structure of Spy, since the differences in the $^{13}C_\alpha$ and $^{13}C_\beta$ secondary chemical shifts between Spy$_{29-124}$ and Spy$_{1-124}$ are small (Supplementary Fig. 8d). In the second approach, we obtained the CSPs of Spy residues by comparing the 2D [$^{15}$N, $^1$H]-TROSY spectra of Spy$_{1-124\ CI}$ and Spy$_{1-124\ CI\ Q25R\ D26K}$ (Fig. 5c, f, i). As mentioned earlier, the Q25R

D26K mutation weakens the interactions between Spy and its N terminus (Fig. 3c). Based on CSPs, we found that these two different approaches revealed very similar residues as potential binding sites of the N terminus (Fig. 5d–f).

Overall, the interaction mode of Spy with Im7$_{A3}$ and its N terminus is similar (Fig. 5d–i), well supporting our hypothesis that Spy uses at least partially overlapping interfaces to bind the client and its N terminus. Together with the biochemical and biophysical characterization described above, this result suggests that the N terminus can directly compete with the client for binding to Spy, which is the basis for promoting client release.

**Tethering of the N terminus enables efficient competition.** In our previous activity assay, the inhibition of Spy$_{29-124}$ activity by the Nt-pep was detected only when the concentration of Nt-pep was at least 14-fold higher of Spy (Fig. 2f). In the native Spy protein, the N terminus is present in a 1:1 ratio with Spy$_{29-124}$ and does not require an excess to function. Apparently, free Nt-pep is

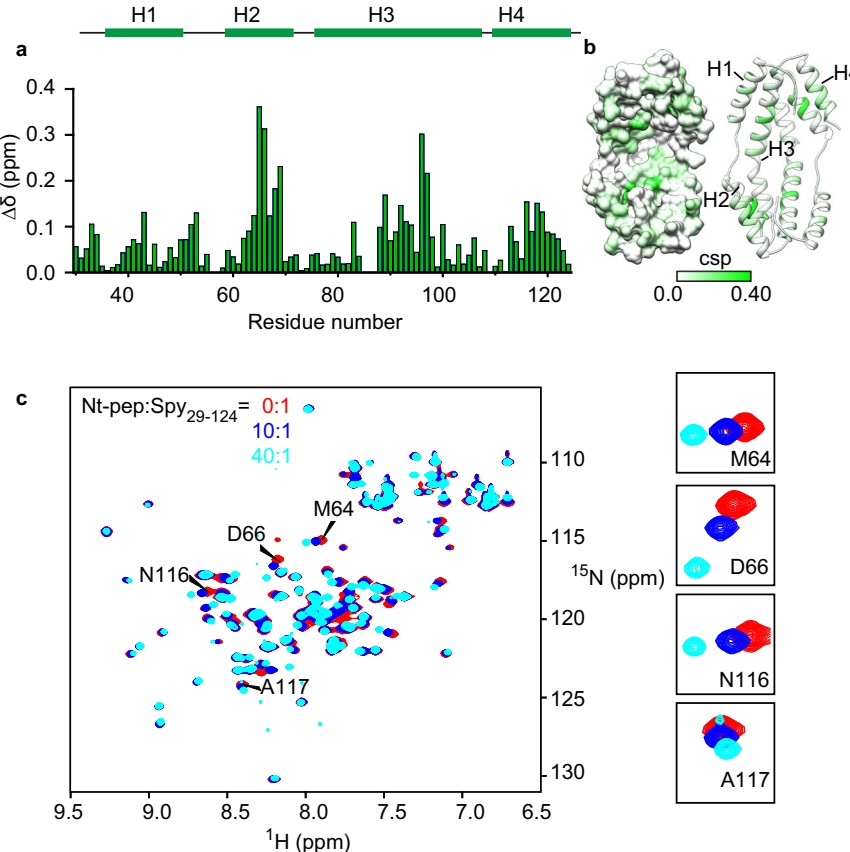

**Fig. 6 Conformational constraints enable the N terminus to facilitate client release. a** CSPs of amide moieties of 100 μM [$^{13}$C, $^{15}$N]-labeled Spy$_{29-124}$ upon binding with 4 mM unlabeled Nt-pep were determined and plotted against the residue number of Spy$_{29-124}$. The secondary structure of Spy$_{29-124}$ is indicated on top. **b** Structural representation of Spy$_{29-124}$ (PBD: 3O39), colored according to the CSP values corresponding to **a** in a white-to-green scale. Color intensities correlate with CSP values, with darker colors indicating larger values. **c** An overlay of the 2D [$^{15}$N,$^{1}$H]-HSQC spectra of 100 μM [$^{15}$N]-labeled Spy$_{29-124}$ in the absence (red) and presence of Nt-pep at molar ratios (Nt-pep: Spy$_{29-124}$) of 10:1 (blue) and 40:1 (cyan). Representative residues with remarkable CSPs including M64, D66, N116, and A117 are labeled and their resonances are highlighted on the right. Source data are provided as a Source Data file.

much less competent in promoting client release compared to the native N terminus of Spy.

We reasoned that tethering the N terminus to the structured region of Spy not only increases the effective local concentration of the N terminus but also imposes conformational constraints on its movement, making it easier to reach the central cavity of the concave surface and thus more effective to compete with the client. Nt-pep in solution lacks such conformational constraints, and thus it may adopt heterologous poses when interacting with the structured region of Spy, which is less effective for the competition. Indeed, we found that the interaction pattern of Nt-pep with Spy$_{29-124}$ is different compared to the fused N terminus (Figs. 5e, f, h, i and 6a, b). The CSPs of Spy$_{29-124}$ reveal that the binding interface of Nt-pep mainly involves the C-terminal regions of helix 2 and helix 3. However, helix 1 and the linker between helix 1 and helix 2 were less affected by Nt-pep, although they were most strongly affected by client binding (Figs. 5d, g and 6a, b).

Titration with increasing concentrations of Nt-pep allowed us to obtain the $K_d$ between Spy$_{29-124}$ and Nt-pep, which was estimated to be 4.7 mM (Fig. 6c and Supplementary Fig. 9). The $K_d$ is even larger when D26 was substituted by arginine or lysine (Supplementary Fig. 9). Therefore, we conclude that Nt-pep can only weakly bind to the structured region of Spy and occupies a binding region that is very different from that of the clients, hindering its capacity to facilitate client release.

## Discussion

Knowing how chaperones bind and release client proteins is fundamental to a full understanding of chaperone action. Chaperones typically recognize and bind hydrophobic amino acid stretches exposed in nascent polypeptides, protein-folding intermediates, and denatured or misfolded proteins[24,25]. Electrostatic interactions also contribute substantially in client association[26–28]. During client protein release, ATP-dependent chaperones, such as GroEL/ES and DnaK/J, regulate the affinity for clients through cycles of nucleotide binding and hydrolysis, as well as co-chaperone association and dissociation, thereby actively releasing client proteins at specific stages of their chaperone working cycles[29]. It is generally believed that ATP-independent chaperones release their client proteins in a passive manner, depending on conformational changes induced by stressors or decreases in chaperone concentration due to protein turnover or cell growth[13,21].

Our investigation of the effect of the disordered termini on Spy's chaperone action revealed an intrinsic client-release mechanism for ATP-independent chaperones. Specifically, the disordered N terminus of Spy forms electrostatic interactions with the client-binding cavity of Spy, thereby promoting client release in a competitive manner (Fig. 7). Interestingly, these intramolecular interactions do not obviously affect the client association rate ($k_{on}$). A plausible explanation for this seemingly counterintuitive observation is that upon binding to the Spy

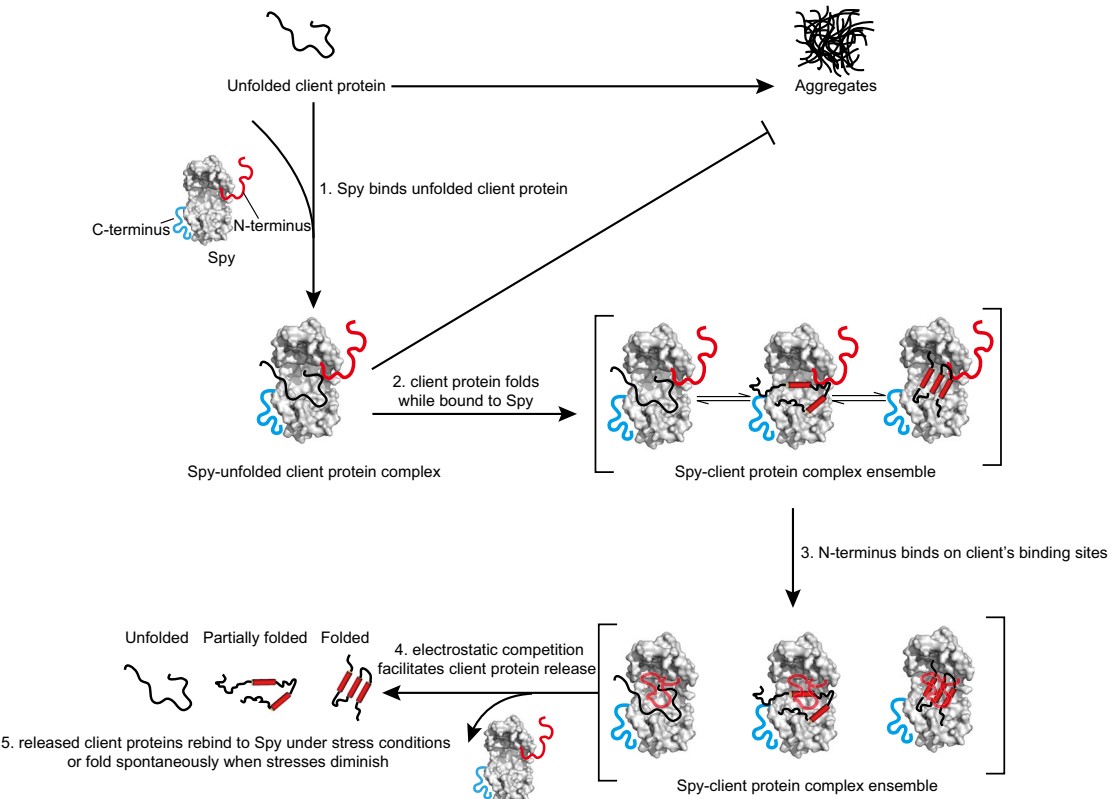

**Fig. 7 Model for the auto-regulated chaperone mechanism of Spy.** Spy binds unfolded or partially unfolded client proteins to suppress their aggregation. Client protein interconverts between different folding states by a previously proposed "folding-while-bound" mechanism[21]. The N terminus of Spy can competitively bind to Spy's concave surface via electrostatic interactions. This process interferes with client retention and thus facilitates client protein release. Released client proteins can rebind to Spy under stress conditions or fold spontaneously when stress conditions diminish. For clarity, only one copy of the N and C termini are shown on the structure of the Spy dimer.

cavity, the N terminus not only neutralizes some of the positive charges on the cavity through residues D2, D10, and D26, but also provides additional positively charged residues (K12, H16, H17, K18, and K20) to attract negatively charged clients in the distance. These contradictory effects may cancel each other out, rendering the net effect of the N terminus on the $k_{on}$ rate negligible. Alternatively, since Spy has been previously reported to change conformation slightly upon client binding[19], the N terminus may be positioned differently upon client binding in order to compete more efficiently with the client for Spy binding. If this is the case, the $k_{on}$ rate is not necessarily affected by the N terminus before client binding. We envision that establishing high-resolution IDR structure determination and simulation methods will help further elucidate the mechanisms involved in the future.

In Supplementary Fig. 2, we observed that the client release-facilitating effect of the N terminus holds for both folded and unfolded Im7 variants. Considering the apparent conformational differences between these two clients, one would assume that they bind to different regions on Spy. However, by analyzing the binding patterns of Im7$_{A3}$ (unfolded) and Im7$_{WT}$ (folded) on Spy$_{1-124}$ by CSP experiments, we found that these clients actually bind a very similar surface on Spy$_{1-124}$ (Supplementary Fig. 10). This observation can be explained by the previously proposed "folding-while-bound" mechanism of Spy, whereby the moderate affinities between Spy and clients allow clients in various folding states to make conformational interconversions upon binding to Spy[21]. Therefore, folded and unfolded clients may end up forming similar conformational ensembles on the Spy surface, leading to the observation of similar binding patterns on Spy. Thus, based on our findings in this study and previous results, we

propose that the mechanism of action by which the N-terminus of Spy facilitates client release may be generally applicable to clients in different folding states (Fig. 7). This property of Spy may offer kinetic advantages allowing it to rapidly cycle between various folding species of client proteins when cells encounter stress conditions and enable spontaneously folding of the released clients when stress conditions diminish (Fig. 7).

Chaperones need to release client proteins at the proper time and the correct cellular location. Persistent binding of client proteins will result in collapsed proteostasis. For example, tenacious binding of Hsp70 by unfolded client proteins limits the amount of Hsp70 used to sequester the key transcriptional regulator, Heat shock factor 1 (Hsf1), leading to collapsed Hsf1 activity regulation, which is presumably associated with transcriptional activation of cancer-specific genes in malignant cells[30,31]. In addition, a previous study reported that Spy variants severely deficient in client release negatively impacted bacterial fitness, leading to prolonged filamentous cell morphology[32].

To evaluate the physiological impact of N terminus removal, we performed a growth assay and found that overexpression of the Spy$_{29-138}$ variant in a Δspy strain resulted in a decrease in strain fitness when cells were recovered from 1% butanol stress (Supplementary Fig. 11a–c). Similar growth retardation was observed in cells expressing Spy$_{D26K}$, but not in cells expressing Spy$_{D26E}$. It has been shown that the folding rate of clients bound to Spy is much slower compared to their spontaneous folding in solution[21,32,33]. Consistent with this, we observed a similar inhibitory effect of Spy on the folding rate of another client, α-LA, and noted that the variant Spy$_{29-138}$ retarded α-LA folding more severely than the Spy wild type (Supplementary Fig. 11d). Thus,

we speculate that the observed decrease in cell fitness could be related to the prolonged binding of Spy$_{29-138}$ to clients, which hinders rapid spontaneous refolding of clients upon stresses decay. Approximately 97% (224 out of 231) of the Spy protein from 54 species have a disordered N terminus, and the three negatively charged residues (D2/E2, D10/E10, and D26/E26) on the N terminus are highly conserved (Supplementary Fig. 12). Therefore, the presence of an N terminus carrying D/E residues and its facilitation of client release may represent an evolutionarily beneficial trait.

Recently, a similar mechanism of client release has been reported for the RNA chaperone Hfq, whose negatively charged, disordered C terminus forms intramolecular electrostatic contacts with its arginine-rich, client-binding Sm domain, thereby facilitating the release of annealed RNAs from Hfq[34,35]. In addition to intramolecular interactions, intermolecular interactions between chaperones and IDRs of their cognate co-chaperones also facilitate client release. For instance, the disordered N terminus of the nucleotide exchange factor Fes1 can mimic the client protein and bind to the substrate-binding domain of Hsp70 to promote client release[36]. Intermolecular interactions are also thought to regulate the chaperone function of small heat shock proteins (sHsps): client binding is partially counteracted by interactions between the disordered C terminus of one sHsp monomer and the client-binding surface of another monomer[37]. Thus, these studies suggest that there may be a generic mechanism that is used by IDRs within chaperones to regulate client binding and release actively.

Interestingly, the peptide derived from the N terminus of Spy (Nt-pep) has much less ability to facilitate client release (Figs. 2f, g and 6 and Supplementary Fig. 9). While this may help explain why Spy homologs have chosen to maintain disordered N termini during evolution rather than develop co-chaperones with N terminus-like properties, it also raises more fundamental questions in chaperone biology that need to be addressed. For example, what are the minimum requirements for a prototypical chaperone that performs holding and folding activities in an ATP-independent manner? How can this knowledge guide the de novo design of artificial chaperones for a variety of cellular tasks? We believe that a deeper characterization of other simple chaperone systems will help to answer these questions.

In summary, our work reveals Spy's active and competitive client release mechanism, which is an example of an emerging paradigm of autoregulation by client-mimicking peptides. This work expands our current knowledge of the functions of IDRs in molecular chaperones. It also highlights the necessity to explore the divergent roles of IDRs in chaperone action to gain a more comprehensive understanding of complex chaperone mechanisms.

## Methods

**Mutagenesis and protein purification**. QuikChange site-directed mutagenesis (Agilent, catalog no. 200518) was employed to generate mutations in *spy* and *ceiE7* (the gene encoding Im7). Spy wild type and mutant proteins were purified as below. Briefly, Spy or Spy variants containing the N-terminal His$_6$-SUMO tag were overexpressed in BL21 (DE3) and initially purified by immobilized nickel-nitrilotriacetic acid (Ni-NTA) affinity chromatography. The His$_6$-SUMO tag was then removed by overnight digestion with ubiquitin-like-specific protease 1. The tag-free Spy and Spy variants were further purified by cation-exchange chromatography and size-exclusion chromatography. Im7 and Im7 variants were purified by a similar procedure to Spy. Protein concentrations were determined by measuring absorbance at 280 nm, and all proteins were aliquoted into 1.5-ml Eppendorf tubes and stored at −80 °C until usage.

**Isotope labeling**. We obtained [$^{15}$N]-labeled Spy variants by growing cells expression the corresponding Spy variants in M9 minimal media supplemented with $^{15}$NH$_4$Cl. Additionally, we obtained [$^2$H, $^{13}$C, $^{15}$N]-labeled Spy variants by growing cells in M9 minimal media containing $^{15}$NH$_4$Cl, $^{13}$C glucose and D$_2$O. Isotopes were purchased from Cambridge Isotope Laboratories, Inc. (Tewksbury, Massachusetts).

**Chaperone activity assay**. Anti-aggregation activities of Spy and Spy variants were determined by detecting their abilities to inhibit the aggregation of DTT-reduced bovine α-LA, type III (Sigma-Aldrich). Briefly, 50 μM α-LA was induced to aggregate by adding 20 mM DTT in a buffer (pH 6.9) containing 50 mM sodium phosphate, 100 mM sodium chloride, and 5 mM EDTA. The light scattering signals of α-LA aggregation were recorded at 320 nm in the absence or presence of Spy and Spy variants using a Synergy HTX Multi-Mode Microplate Reader for 10 h at room temperature. To quantify the activities of Spy variants relative to the wild type Spy, a standard curve was generated by plotting the slopes of α-LA aggregation curves (from 180 to 240 min) against Spy wild type: α-LA ratios (from 0.0375 to 2). The standard curve was fitted with Origin 8.6 software using a one-phase exponential decay equation and the relative activities of Spy variants to the wild type from 3 independent experiments were calculated according to the standard curve.

**Biolayer interferometry**. The association and dissociation kinetics of Spy variants toward two fully unfolded clients (carboxymethylated α-LA and Im7A$_3$), one partially folded client (Im7$_{AAW}$), and one folded client (Im7 wild type) were measured with an Octet RED96 system (ForteBio, Menlo Park, CA). Each client protein was biotinylated with EZ-link NHS-biotin (Thermo Fisher Scientific, Waltham, Massachusetts) at a 1:1 molar ratio of biotin to protein before loading onto the streptavidin sensors. Then each client-loaded sensor was immersed into a buffer (40 mM HEPES and 150 mM NaCl, pH 7.5) containing different concentrations of Spy and Spy variants (0.25, 0.5, 1, 1.5, and 2 μM) for chaperone-client complex binding. Afterward, each sensor was transferred to the buffer without Spy to dissociate the chaperone-client complexes. Kinetic parameters were obtained by fitting the association and dissociation curves in SigmaPlot 12.0 software according to the following equations, respectively:

$$y = a \times \left(1 - e^{-k_{obs} \times t}\right) + y_0 \times t \tag{1}$$

$$y = a \times e^{-k_{off} \times t} + y_0 \times t \tag{2}$$

where $y$ is the observed wavelength shift, $y_0$ is the constant to correct for baseline drift, $a$ is the maximal wavelength shift, $k_{obs}$ is the observed association rate constant, $k_{off}$ is the dissociation rate constant, respectively. The association rate constant ($k_{on}$) is then obtained according to the following equation:

$$k_{obs} = k_{on} \times [\text{Spy}] + k_{off} \tag{3}$$

where [Spy] is the Spy concentration. The dissociation constant $K_d$ is calculated according to:

$$K_d = k_{off}/k_{on} \tag{4}$$

**NMR spectroscopy**. The NMR experiments were carried out at 25 °C on an Agilent DD2 600 MHz or 800 MHz spectrometers equipped with cryogenic probes. The U-[$^2$H, $^{13}$C, $^{15}$N] NMR samples were prepared in NMR buffer A containing 40 mM HEPES-NaOH (pH 7.5), 150 mM NaCl, 100×cocktail protease inhibitor, 1 mM PMSF, 0.03 % NaN$_3$ (w/v), 3 mM DTT, and 90% H$_2$O/10 % D$_2$O. Chemical shifts were referenced to an external 4,4-dimethyl-4-silapentane-1-sulfonic acid. The backbone assignments were obtained using the TROSY versions of standard triple resonance experiments, including HNCO, HN(CA)CO, HNCA, HN(CO)CA, HN(COCA)CB and HNCACB[38]. In addition, a nuclear overhauser effect (NOE) mixing time of 120 ms was recorded for a 3D $^{15}$N-edited NOESY-HSQC experiment to validate backbone assignments. All spectra employed a non-uniform sampling scheme in the indirect dimensions and were reconstructed by the multi-dimensional decomposition software MddNMR[39,40] interfaced with NMRPipe[41]. The spectra were analyzed with NMRFAM-SPARKY[42]. ΔCα and ΔCβ values were calculated by subtracting the random coil chemical shifts[43] from experimentally determined.

To investigate the interaction between Spy and each client protein, 0.4 mM U-[$^2$H, $^{13}$C, $^{15}$N] Spy$_{1-124}$, U-[$^2$H, $^{13}$C, $^{15}$N] Spy$_{29-124}$ and U-[$^{13}$C, $^{15}$N] Spy$_{29-124}$ were titrated with unlabeled Im7 or Im7$_{A3}$ at a 1:1 molar ratio in NMR buffer A. The 2D $^1$H-$^{15}$N TROSY spectra were recorded at 25 °C. The chemical shift was calculated using the equation:

$$\text{CSP} = \sqrt{(\Delta H)^2 + (0.2\Delta N)^2} \tag{5}$$

**Nt-pep NMR titration experiments**. Titration experiments were performed on the Agilent DD2 800 MHz equipped with a cryogenic probe by mixing U-[$^{15}$N] Spy$_{29-124}$ and different concentrations of unlabeled peptides (Nt-pep, Nt-pep D26K and Nt-pep D26R). The concentration of U-[$^{15}$N] Spy$_{29-124}$ was maintained at 0.1 mM. The 2D $^1$H-$^{15}$N HSQC spectra were recorded for seven molar ratios (1:2, 1:6, 1:10, 1:14, 1:20, 1:30, 1:40) of Spy$_{29-124}$: peptide in NMR buffer B (40 mM HEPES-NaOH and 150 mM NaCl, pH 7.5) at 25 °C. The dissociation constants for residues shown in Supplementary Fig. 9 were determined by fitting the titration curves to the one-site binding model[44] using the equation:

$$\Delta_{obs} = \Delta_{max} \times \frac{(K_d + P_0 + L_0) - \sqrt{(K_d + P_0 + L_0)^2 - 4P_0 L_0}}{2P_0} \tag{6}$$

where $\Delta_{obs}$ is the observed CSP value, $\Delta_{max}$ is the maximum CSP value, $K_d$ is the

equilibrium dissociation constant, $P_0$ and $L_0$ indicate the total concentrations of the unlabeled $^{15}$N-labeled Spy$_{29-124}$ and Nt-pep, respectively.

**Spin labeling and PRE experiments**. Spin labeling of the cysteine mutants of different Spy variants with MTSL was performed according to a published protocol[45], which is described as follows. First, upon reducing the cysteines within the Spy mutants, DTT was removed from the samples by buffer exchange into 40 mM HEPES-NaOH (pH 7.5), 150 mM NaCl. The protein samples were then diluted to the concentration of 20 μM and reacted with a 20-fold excess of MTSL. Labeling was performed for 12 h on a rocking shaker at room temperature. The unreacted MTSL was removed by ultrafiltration. Labeling efficiency was measured using matrix assisted laser desorption ionization-time of flight mass spectrometry (MALDI-TOF MS) and was above 80% for all samples. The paramagnetic NMR sample was reduced by adding 5-fold excess of ascorbic acid. $^{1}$H$-^{15}$N TROSY-HSQC experiments were performed at 25 °C on a Bruker Avance 600 MHz spectrometer for both paramagnetic and diamagnetic samples. The PRE effect is quantified by the ratio of each individual peak between the diamagnetic and paramagnetic states of the same protein. Spectra were processed and analyzed with Sparky and NMRPipe[41].

**Modeling of disordered N terminus**. Since the N termini is missing in the crystal structure of Spy (PDB: 3O39), we have to rebuild it and attach it to the crystal structure. Four N termini with different conformations were constructed. The first one was generated simply using CABS-dock[46]. The other three were constructed using Amber18[47] according to an in-house developed protocol, as detailed below.

First, 5000 random coil conformations were generated using OOPS and SCWRL4[48] and then linked to both monomers of the crystal structure of Spy. The ff14SB force field[49] was used for the structural refinement. Each structural model was solvated in a truncated octahedral SPC/E water box and neutralized using the chloride ions. The solvated system was subjected to energy minimization for 1000 steps with harmonic restraints applied on the heavy atoms of the Spy's cavity regions (force constant 500 kcal mol$^{-1}$ Å$^{-2}$). The optimized system was first heated from 0 to 1000 K and then cooled from 1000 to 0 K under NVT ensemble with Berendsen thermostat. Both heating and cooling were divided into five stages with a temperature increment of 200 K. During this simulated annealing procedure, simulation was performed for 40 ps (heating) or 120 ps (cooling) in total with harmonic restrains (force constant 500 kcal mol$^{-1}$ Å$^{-2}$) applied on all heavy atoms except for the residues of N terminus. Finally, the structure was minimized for 1000 steps under implicit solvent (igb = 5) in the absence of restraints and the energy was reported. In the resulting structure pool of Spy with N termini, we filtered out those with D26 located inside the cavity of Spy to make sure D26 will not by accidentally "stick to" the positively charged residues in the cavity and thus cannot escape from the cavity within the reasonable simulation time. In doing so, an upper plane of Spy's cavity was determined by four points (the geometry center of the Cα atoms of E110 and K112, and the geometry center of the Cα atoms of E120 and K121, with both centers calculated from each monomer). The top 3 structures with lowest energy were selected for final MD simulations.

**MD simulations**. A total of four conformations were used as the template to build the Spy variants. These variants were modeled so that each Spy monomer has a cysteine insertion before the D26 residue and one or two mutations to the predicted D26-interacting residues, including K54D R55D, R61D R62D, and R89D, as well as the K75D that serves as a control. To model the PRE spin label, an MTSL molecule was attached to the N-terminal cysteine residue. The resulting proteins were simulated with the ff14SB force field. The force field parameters of the MTSL tag were generated using the CHAMBER module of Amber18[50]. The proteins were solvated in a truncated octahedral box of SPC/E water molecules such that the boundary of water box was at least 10.5 Å away from any atoms of the protein. Sodium and chloride ions were then added to achieve electroneutrality of the system and an ionic strength of 150 mM NaCl. The hydrogen mass repartition scheme was applied to enable the use of a larger integration step (4 fs) in order to achieve higher production rate of MD, by using the ParmEd module of AMBER18 to generate modified topology files for subsequent MD simulations. The solvated system was minimized for 1000 steps with harmonic restraints (force constant 500 kcal mol$^{-1}$ Å$^{-2}$) on Spy's structured region, followed by 1000 steps without restraints. The system was then heated for 14 ps from 0 to 310 K, and equilibrated for 1 ns at 310 K. This temperature (rather than the experimental temperature 298 K) was chosen to increase the conformational sampling. The 1 μs production-stage of MD simulation was conducted at 310 K using the NPT ensemble. During the simulations, all bonds involving hydrogen atoms were constrained using the SHAKE algorithm implemented in Amber18. The nonbonded cutoff was set to 10 Å. Both the cpptraj module of Amber18 and our in-house Python scripts (using Python 2.7) were used for analyzing the distance distribution. VMD 1.9.3[51] and PyMOL (The PyMOL Molecular Graphics System, Version 2.1.1 Schrödinger, LLC.) were used for visualizing the MD results.

**Reporting summary**. Further information on research design is available in the Nature Research Reporting Summary linked to this article.

## Data availability

The backbone chemical shifts generated in this study have been deposited in the BioMagResBank under the accession code BMRB 51107 ($^{1}$H, $^{15}$N and $^{13}$C backbone chemical shift assignment of Spy$_{29-124}$), BMRB 51108 ($^{1}$H, $^{15}$N and $^{13}$C backbone chemical shift assignment of Spy$_{1-124}$) and BMRB 51109 ($^{1}$H, $^{15}$N and $^{13}$C backbone chemical shift assignment of Spy$_{29-124}$ in complex with Im7$_{A3}$). The previously published crystal structure of Spy used was obtained from the Protein Data Bank: 3O39. Source data are provided with this paper.

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

## Acknowledgements

We thank James C. A. Bardwell (University of Michigan) and Bikash Ranjan Sahoo (University of Michigan) for helpful suggestions and insightful discussion. We thank Dr. Ning Xu in the BioNMR facility of the China National Center for Protein Sciences Beijing, for providing facility assistance. This work was supported by the National Natural Science Foundation of China (NSFC) grants 31661143021 and 31400664 (to S.Q.), funds from the Tsinghua-Peking Joint Center for Life Sciences and Beijing Advanced Innovation Center for Structural Biology (to Y.X.), the Fundamental Research Funds for the Central Universities (22221818014 to J.X.), the Research Program of State Key Laboratory of Bioreactor Engineering (to S.Q.), and the Shanghai Frontier Science Center of Optogenetic Techniques for Cell Metabolism Shanghai Municipal Education Commission, grant 2021 Sci & Tech 03 28 (to S.Q.).

## Author contributions

W.H. and X.L. contributed equally to this work. W.H., B.W., Y.X., and S.Q. conceptualized project. W.H., Y.Y., J.M., L.B., and J.Z. performed biochemical experiments, analyzed the data, and prepared NMR samples. X.L. and H.X. collected and analyzed the NMR data. X.L. performed MD simulations and analyzed the data. W.H., X.L., J.X., B.W., Y.X., and S.Q. wrote the paper with input from all authors. All authors read and approved the manuscript.

## Competing interests

The authors declare no competing interests.
