## [Peer Review File · Nature Communications]

REVIEWER COMMENTS

Reviewer #1 (Remarks to the Author):

In this manuscript He et al present a very comprehensive characterization of the role that the N-terminus of Spy plays in regulating the affinity that this molecular chaperone has for its clients and, therefore, its activity. The authors reach the conclusion that the N-terminus interacts with the client binding cleft mainly via a specific residue (Asp 26) and that this ensures the efficient release of substrates. The characterization is particularly comprehensive and there are interesting aspects in this work. The conceptual novelty is limited, however, particularly in the absence of a compelling functional validation, and there are a few points where the experimental or simulation procedures should be improved. Once these concerns have been addressed the authors should resubmit the paper but to a more specialized journal such as JMB.

Major issues

- 1) In the abstract the authors state that their work challenges the notion that ATP-independent chaperones such as Spy 'passively release' their clients. This statement should be removed from the abstract because it implies that Spy 'actively releases' its clients, which it does not. In the mechanism proposed by the authors the release is still passive i.e. does not imply energy consumption.
- 2) The experiments reported in Figure 1c,d,e show how removal of residues 1 to 24 from the N-terminus decreases the activity of Spy because it accelerates substrate release and as a consequence decreases affinity. Since in the mechanism put forward by the authors the role of the N-terminus is to compete against the substrate the outcome of these experiments contradicts this conclusion.
- 3) The experiments are carried out with care (although see below) and the differences measured in the affinities and rates are likely correct. These differences are however small, typically dividing or multiplying the values of the full length molecular chaperone by two, and one wonders whether these are differences, although real, are functionally relevant. This is despite the experiments presented in Figure S9, the conclusions of which are too speculative.
- 4) The simulation work discussed in lines 175 to 179 is of little use. Visualization of the trajectories shows that the N-terminal regions hardly change structure, indicating that their behavior is driven by the (arbitrary) choice of starting conformation. They do not anything to the manuscript and should therefore be removed, unless the authors decide to extend them, carry out a careful analysis of the

convergence of the trajectories and ensure the results are robust. The same applies to later work on the interpretation of the PRE data.

5) The results presented in Figure 4 are quite disconcerting. The introduction of mutations to Asp in the positively charged residues of the cleft in the globular domain of Spy should decrease its affinity for the N-terminal region and yet in all mutants except one (R89D) the authors obtain a change in the PREs that involves some shortened distances that are hardly discussed in the manuscript.

Minor issues

1) From a technical perspective the high similarity of the values of k_D (Figure 1e) and K_D (Figure 1d) reported for the truncation mutants starting at position 27 (blue) and 29 (purple) is problematic. Would the technique used for these experiments differentiate the rates of affinities if they would indeed be different? In other words is this biophysical technique well-suited to differentiate affinities in this range?

2) The concentration of Spy used in the experiment shown in Figure 2g should be specified in the figure and the caption.

3) The interpretation of Figures 5 and 6 would be greatly facilitated with a scheme illustrating the relative position of helices 1 to 4 in the structure of Spy.

4) Two observations suggest that although electrostatic interactions play an important role there are other interactions between the N-terminal region and the cleft: mutation of residues 25 and 26 to R and K does not lead to an NMR spectrum indicating lack of interaction (Fig. 5c,f) and titration of the protein devoid of the N-terminus with N-terminal peptide with position Asp 26 mutated to R or K slightly weakens the interaction (Fig. S7) but does not abolish it. The authors should discuss these results explicitly and provide possible explanations for them.

Reviewer #2 (Remarks to the Author):

In this manuscript the teams of Shu Quan and Yi Xue investigate how intrinsically disordered regions of chaperone proteins affect their function, building on kinetic and structural information. The work focuses on the periplasmic protein chaperone Spy and the step of substrate release. Given that Spy is ATP-independent, the mechanistic impact of this work is very high, as in the absence of a coupled ATP hydrolysis cycle, regulation of the progression through different steps of cycle is not apparent. Therefore, detailed dynamic/thermodynamic/structural studies are required to understand the steps of substrate entry and release from the chaperone.

The authors utilize a broad array of techniques to obtain mechanistic understanding of substrate binding and release from Spy including: (i) a functional -folding- assay, (ii) NMR spectroscopy (with PREs), which is the most appropriate approach to capture the dynamic interaction of the N-terminus with the concave surface of Spy, (iii) a very large number of mutants (deletion constructs, plus multiple point mutants), (iii) reciprocal mutation of charged residues in the N-terminus and the concave, substrate binding, surface of Spy (that's the most elegant part of the work that demonstrates very nicely the proposed mechanism) and (iv) data with the N-terminal peptide.

The manuscript is very well written and all the conclusions are nicely supported.

Overall, this is a very significant piece of work in the field of chaperone biology and protein folding, which deserves publication at Nature Communications. Minor revisions are suggested below.

Minor comments:

1) Line 201: substitute has for have

2) The data presented in Fig. 5b, e, h make a better fit with the data presented in Fig. 3. Instead, it seems that the authors have segregated data based on the approach utilized (PREs vs. CSP). Still the flow of the text does make sense as presented now. Looking at the PREs (Fig 3) my first thought was "what if the sticky MTSL binds non-specifically to the client binding site?" and "why simple spectrum overlays of the different constructs is not presented in addition to PREs to clarify this?". Then, in Fig. 5 I see the data...

3) The authors may want to present data (HSQC overlays or functional assays or kinetic measurements) acquired at different (higher) ionic strength. This excludes the possibility of affecting the structure of Spy upon any of the modifications utilized in here.

4) The last paragraph -as is- does not add anything to the manuscript. On the contrary, it makes it weaker. Of course, tethering the N-terminus to the main body of Spy has a tremendous functional impact just because of the "tethering" effect, what the authors describe as "increases the effective local concentration". Obviously, this imposes particular conformational constraints that affect the interaction compared to observations made with the peptide added in trans.

Looking at Fig. 5e, the CSP for H1, the H1-H2 loop and H4 are comparable to that shown in Fig. 6a, while the large CSP for residues before 35 can still be attributed to the tethering effect. The large CSP shown in Fig. 6a for other regions may just be the result of 40x excess, where non-specific binding may appear having a similar effect (or even more pronounced) as a weak, but specific binding.

Reviewer #3 (Remarks to the Author):

In the manuscript entitled “An Active client protein release mechanism of an ATP-independent chaperone”, He et al. show that the N-terminal disordered sequence of the periplasmic, ATP-independent chaperone Spy facilitates client release. By using a combination of kinetic measurements, NMR spectroscopy, and molecular dynamics simulations, the authors studied the interactions between Spy (and several mutants) with their substrates and observed that the disordered N-terminal sequence of Spy facilitates substrate release from its positively charged, concave substrate-binding region. The authors propose that this release operates mainly via a “dynamic competition” mechanism mediated by electrostatic interactions, and speculate that this type of “release” can be a general mechanism used by ATP-independent chaperones to actively release their bound substrates.

I enjoyed reading this manuscript, and I think that the reported findings are novel. The experiments are clever and rigorously performed. However, the way some results are interpreted is confusing. I have a main “question” related to this that would need to be addressed in order to clarify the interpretation of the results (see below).

Major point

The authors show that the N-terminal sequence of Spy can, because its electrostatic interactions with the “positively charged” concave cavity that binds substrates, compete off the bound substrates. This means that the effect of the N-terminus on substrate release operates via a “conformational selection” by binding to the available surface when the substrate is NOT bound to the cavity. If this was the case, then the effect of the N-terminus on substrate binding should be observed in BOTH, the k_{on} and the k_{off} for substrate binding (not ONLY k_{off} as observed).

Moreover, the authors propose, based on the molecular simulations, that the N-terminal sequence is “attracted” by the positively charged surface that binds the substrates. The authors perform these simulations using the “empty” Spy. Again, if this effect can be seen in the substrate-free Spy, the competition effect should be observed both in k_{on} and the k_{off} .

Because the “free” N-terminus sequence added in trans as a peptide does NOT have the same effect as the N-terminus in the full length Spy, the authors propose that “the tethering of the N-terminus to the structured region of Spy not only increases the effective local concentration of the N-terminus but also imposes conformational constraints on its movement, making it easier to reach the central cavity of the concave surface and thus, more effective to compete with the client”. I was very pleased to read this (although at the very end of the results, this result should be moved up in the results section!) but I am not convinced by the interpretation: If the sequence of the N-terminal sequence in the context of the full length Spy binds the substrate-binding cavity and compete off the

substrate, again, this effect should be also observed in the k_{on} of the substrate (in the presence AND the absence of the N-terminus in Spy).

An alternative interpretation could be that, when substrate is BOUND, the chaperone undergoes a conformational change that makes the N-terminus sequence more effective in competing off the substrate (restricts motion orienting it optimally?). In this case, the N-terminus would not very have an effect on k_{on} . But after substrate binding, the N-terminus could be differently positioned, being now able to effectively compete off the bound sequence.

Can you show this by NMR? Is substrate-bound Spy amenable for NMR studies?

Minor points

1. Can you describe a bit more the “super Spy” variant created previously?
2. Figures: I very much appreciate that the authors show raw data for the preformed experiments. In some cases (for sure for Figure 1) the raw results could be moved to the supplementary material for clarity.
3. Please make the labels of the Y axis a bit larger.
4. Figure 3: not need to specify “180°” for each graph if they are all the same. Same with “residue number” labels (text and numbers). That would help with clutter (same in Fig 4).

REVIEWER COMMENTS

Reviewer #1 (Remarks to the Author):

In this manuscript He et al present a very comprehensive characterization of the role that the N terminus of Spy plays in regulating the affinity that this molecular chaperone has for its clients and, therefore, its activity. The authors reach the conclusion that the N terminus interacts with the client binding cleft mainly via a specific residue (Asp 26) and that this ensures the efficient release of substrates. The characterization is particularly comprehensive and there are interesting aspects in this work. The conceptual novelty is limited, however, particularly in the absence of a compelling functional validation, and there are a few points where the experimental or simulation procedures should be improved. Once these concerns have been addressed the authors should resubmit the paper but to a more specialized journal such as JMB.

Our remarks to reviewer 1: Thank you for your careful review of our manuscript. Below, I respond to all your comments on a point-by-point basis.

Major issues

1) In the abstract the authors state that their work challenges the notion that ATP-independent chaperones such as Spy passively release their clients. This statement should be removed from the abstract because it implies that Spy actively releases its clients, which it does not. In the mechanism proposed by the authors the release is still passive i.e. does not imply energy consumption.

Our response: We did not mean to imply that Spy actively releases its clients and have revised the abstract to make sure we don't. The new sentence now reads like this:

“Our results reveal a self-aided client release mechanism independent of energy input, thus enriching the current knowledge on how ATP-independent chaperones release their clients and highlighting the importance of synergy between IDRs and structural domains in regulating protein function.”

2) The experiments reported in Figure 1c,d,e show how removal of residues 1 to 24 from the N terminus decreases the activity of Spy because it accelerates substrate release and as a consequence decreases affinity. Since in the mechanism put forward by the authors the role of the N terminus is to compete against the substrate the outcome of these experiments contradicts this conclusion.

Our response: These results do not contradict our main conclusions. Removal of residues 1 to 23

does not abolish the interaction between the rest of the N terminus (including residue D26) and the concave surface of Spy. Because D26 is the key residue that mediates this intramolecular interaction (in Fig. 2d and Fig. 3c we show that replacing residue D26 with lysine or arginine strongly decreases client release rates and greatly reduces the transient contacts between the N terminus and Spy's concave surface), the remaining N terminus (residues 24-28) is still able (indeed, even more capable) to compete with the substrate for binding to Spy, thereby facilitating substrate release.

When the N terminus was further shortened to remove residues 24-26, the competition effect was almost eliminated, consistent with abrupt decreases in substrate release rates and smaller K_d values (Figure 1. d, e). Indeed, it was these unexpected changes in Spy activity when we systematically shortened the N terminus that promoted our study of the roles of residues 24-26 in Figure 2.

3) The experiments are carried out with care (although see below) and the differences measured in the affinities and rates are likely correct. These differences are however small, typically dividing or multiplying the values of the full length molecular chaperone by two, and one wonders whether these are differences, although real, are functionally relevant. This is despite the experiments presented in Figure S9, the conclusions of which are too speculative.

Our response: Thank you for asking this critical question. To our knowledge, 2-3 fold of affinity change in chaperone-client interactions or chaperon-(co)chaperone interactions is sufficient to impact chaperone function.

For example, a point mutation of leucine 42 of SecB to arginine (L42R) decreases its affinity for the client by a factor of 2, leading to defects in the anti-aggregation activity and protein translocation capacity of SecB (Bechtluft et al, *Biochemistry*, 23;49(11):2380-8, 2010). The Wickner group reported that a 3-fold reduction in binding affinity between DnaK and the *E.coli* Hsp90 (Hsp90_{EC}) resulted in defective functional collaboration between these two chaperones in client reactivation (Kravats et al, *J Mol Biol*, 429(6):858-872, 2017). Our previous studies also showed that strengthening the affinity between Spy and client proteins by 1.27 to 5.75-fold resulted in a significant 1.10 to 2.19-fold enhancement of the *in vivo* folding activity of Spy (Quan et al, *eLife*, 3:e01584, 2014; He et al, *J Biol Chem*, 295(42):14488-14500, 2020).

To strengthen our *in vivo* data, we reperformed the cell growth assays and incorporated more Spy variants to draw conclusions. We found that overexpression of the Spy variant Spy_{D26K} (which has a slower k_{off} rate than wild-type Spy) resulted in a decrease in strain fitness, while overexpression of the variant Spy_{D26E} (which has a similar k_{off} rate to wild-type Spy) did not impact cell growth. We have added these additional data to the new Supplementary Fig. 11, and also updated the sentences below to our manuscript on page 19 as follows:

“Similar growth retardation was observed in cells expressing Spy_{D26K}, but not in cells

expressing Spy_{D26E}.”

“Therefore, the presence of an N terminus carrying D/E residues and its facilitation of client release may represent an evolutionarily beneficial trait.”

Supplementary Fig. 11

Supplementary Fig. 11 Deleting the N terminus of Spy impedes client refolding and inhibits *E. coli* growth. (a-b) Growth curves of *E. coli* strains carrying the expression plasmid of Spy wild type or **Spy_{D26E}**, **Spy_{D26K}**, or Spy₂₉₋₁₃₈, or the empty vector, with (a) or without (b) IPTG induction. Overnight cultures of different strains were diluted 100-fold into fresh LB media. After 2 h of growth, the cultures were treated with or without 1 mM IPTG for 2h and then with 1% butanol for another 1.5 h. Then, 0.1 OD cells from these cultures were inoculated into fresh LB media supplied with (a) or without (b) 1 mM IPTG, and cell growth was monitored. Comparisons of OD_{600nm} values for various cultures at t = 12 h are shown to the right (mean \pm SD, n = 4; individual data points are shown; one-way ANOVA with Tukey's multiple comparisons test). (c) For the IPTG-induced cultures in (a), samples were collected at t = 9 h, the periplasmic fractions of the cells were extracted and visualized on SDS-PAGE. Quantification of the band intensities corresponding to Spy wild type or **Spy variants** in arbitrary units (a.u.) is shown to the right (mean \pm SD, n = 4; individual data points are shown; one-way ANOVA with Tukey's multiple comparisons test). (d) Kinetic traces of urea-denatured α -LA (1 μ M) refolded in the absence or presence of 2 μ M Spy wild type or Spy₂₉₋₁₃₈ monitored by intrinsic tryptophan fluorescence. The observed refolding rates ($k_{UN, obs}$) were obtained by fitting the kinetic curves with a single exponential equation. Representative traces of two independent experiments are shown.

4) The simulation work discussed in lines 175 to 179 is of little use. Visualization of the trajectories

shows that the N-terminal regions hardly change structure, indicating that their behavior is driven by the (arbitrary) choice of starting conformation. They do not anything to the manuscript and should therefore be removed, unless the authors decide to extend them, carry out a careful analysis of the convergence of the trajectories and ensure the results are robust. The same applies to later work on the interpretation of the PRE data.

Our response:

Thanks for raising this question. What the reviewer said is correct. Previously, our MD trajectories all started with an initial conformation of N terminus generated simply by CABS-dock, and the convergence issue poses a potential risk. At the reminder of the reviewer, we repeated the MD simulation using a different N-terminal conformation, and the resultant trajectory indeed looked different. We also tested accelerated MD (aMD), but it turned out that the convergence remained inadequate even for aMD. Finally, we chose to record multiple 1- μ s trajectories that started with different initial conformations of N terminus. In brief, we constructed three additional structures featured by different N-terminal conformations (see Methods for details). In doing so, a large pool of random coil N-terminal tails was generated, and were attached to the body of Spy. After energy minimization, the top 3 structures with lowest energy were chosen for MD simulations. Meanwhile, to confirm that the C-terminal tail does not interrupt the interaction between the N terminus and the Spy cavity, we removed C terminus when running these simulations (please note that we used Spy with C terminus truncated for NMR measurements). We also repeated the original MD simulation, but in the absence of C terminus, in order to make a fair comparison with the old result. In all these four trajectories, D26 residue showed a remarkable tendency to stay away from the cavity of Spy (see the new Supplementary Fig. 4, which is also shown below). We performed the similar treatment for the MD simulations in the PRE section. Due to the high computational cost of MD simulations, we chose to evaluate only the robustness of the R89D trajectory. Indeed, we found all the four R89D trajectories, which started with the same set of initial conformations as above, showed clear tendency of pushing the MTSL spin label out of the bottom of the cavity (see the new Supplementary Fig. 7, which is also shown below). We have modified the manuscript accordingly (Pages 14, 24, and 25).

“To evaluate the robustness of these MD results and to confirm that the C-terminal tail does not interrupt the interaction between the N terminus and the Spy cavity, we also performed MD simulations on Spy_{1-124 CI} and Spy_{1-124 CI R89D} by using different starting conformations for the N terminus, all with the C terminus removed (see **Methods**). We found that all the four R89D trajectories showed clear tendency of pushing the MTSL spin label out of the bottom of the cavity (**Supplementary Fig. 7**).”

Supplementary Fig. 4

Supplementary Fig. 4 Distributions of the distance between residue 26 and the cavity of Spy as observed in the MD simulations. The normalized distributions for the Spy₁₋₁₂₄ and Spy₁₋₁₂₄D_{26R} are shown in black and orange, respectively, with average distances indicated as dashed lines in corresponding colors. The intramolecular distance was calculated using the geometry center of all atoms in D26/R26 and the geometry center of the cavity region (residues 29-124) for each monomer, after combining four MD trajectories with different initial conformations.

Supplementary Fig. 7

Supplementary Fig. 7 Distributions of the distance between the MTSL label and the bottom of Spy cavity as observed in the MD simulations. (a) Position of the MTSL spin label on the Spy cavity for MD simulations of Spy₁₋₁₂₄ CI (left) and Spy₁₋₁₂₄ CI R_{89D} (right). The two monomers of Spy are shown as cartoons in white and gray, and the spin labels from different monomers are represented as spheres in cyan and orange, respectively. The statistics was performed using 1000 MD frames uniformly resampled in the combined trajectory from four MD simulations with different initial conformations of N terminus. The nitroxide oxygen (O1) atoms from each monomer are represented by cyan and orange spheres, respectively. Compared to Spy₁₋₁₂₄ CI, the O1 atoms of Spy₁₋₁₂₄ CI R_{89D} evidently moved away from the bottom of the cavity. (b) The normalized distributions of the distance between the MTSL label and the bottom of Spy cavity for the Spy₁₋₁₂₄ CI (black) and Spy₁₋₁₂₄ CI R_{89D} (orange), with average distances indicated as dashed lines in corresponding colors. The intramolecular distance was calculated using the O1 atom of MTSL and the geometry center of all atoms in R89/D89 for each monomer, after combining four MD trajectories with different initial conformations.

5) The results presented in Figure 4 are quite disconcerting. The introduction of mutations to Asp in the positively charged residues of the cleft in the globular domain of Spy should decrease its affinity for the N-terminal region and yet in all mutants except one (R89D) the authors obtain a change in the PREs that involves some shortened distances that are hardly discussed in the manuscript.

Our response: We apologize that we did not make enough discussion in the previous submission. First, we emphasize that not all the positively charged residues located on the concave surface are involved in D26-interaction. Second, D26 may interact with several positively charged residues during a period of time. An Asp substitution to one of such residues may push D26 away from this site and move it to other preferred sites, causing changes in the accessibility of the MTSL spin label towards the concave surface. As D26 is changing its distance distribution on the accessible concave surface, it may get closer towards certain residues while away from other residues. This is the reason why some shortened distances were observed in Figure. 4b.

We have now added the following sentences on page 13:

“Because we saw considerable variation in the distance distribution of the MTSL label, it is possible that D26 interacts with several positively charged residues on the concave surface. The disruption of the interaction between D26 and one such residue may be partially compensated by enhancing the interaction between D26 and another such residue, which is in accordance with the inherent flexibility of the N terminus.”

Minor issues

1) From a technical perspective the high similarity of the values of k_D (Figure 1e) and K_D (Figure 1d) reported for the truncation mutants starting at position 27 (blue) and 29 (purple) is problematic. Would the technique used for these experiments differentiate the rates of affinities if they would indeed be different? In other words, is this biophysical technique well-suited to differentiate affinities in this range?

Our response: Biolayer interferometry (BLI) is a powerful tool for probing protein-protein binding kinetics (Wang et al, Nature, 595(7867):426-431, 2021; Sauer et al, Nat Struct Mol Biol, 28(6):478-486, 2021; Shih et al, Nat Commun, 11(1):5950, 2020) with typical detection range of 10 pM to 1 mM for affinity, 10^{-6} to 10^{-1} s⁻¹ for k_{off} rate, and 10^2 to 10^7 M⁻¹s⁻¹ for k_{on} rate (<http://separations.co.za/wp-content/uploads/2017/05/OCTET.pdf>; <https://paralab-bio.es/wp-content/uploads/2020/03/ParalabBIO-ForteBio-octet-red96-system.pdf>). The kinetic constants in Fig.1 which range from 10^{-1} to 10^0 μM for affinity and 10^{-2} to 10^{-1} s⁻¹ for k_{off} rate are within the detection limits of BLI technique. Previously, we also used this approach to distinguish the different

affinities and k_{off} rates of activity-enhanced Spy variants and obtained values in the same range (Quan et al, *eLife*, 3:e01584, 2014; He et al, *J Biol Chem*, 295(42):14488-14500, 2020). Other researchers also employed this method to differentiate affinities and k_{off} rates of chaperone-client, kinase-ligand, ubiquitin-deubiquitinase in this range (Huang et al, *Nature*, 537(7619):202-206, 2016; Reshetnyak et al, *Nature*, 600(7887):153-157, 2021; Zhang et al, *Nat Chem Biol*, 9(1):51-8, 2013).

To get an independent evaluation of the K_d , we measured the binding affinities of the Spy truncation variants starting at position 27 and 29 towards Im7_{AAW} by intrinsic tryptophan fluorescence titration method. We obtained very similar affinities for these variants (see figure below), indicating that these values are similar in nature. Considering all of these, we believe that BLI technique is suitable for differentiating the affinities of various truncation variants of Spy shown in Fig. 1.

Figure legend: Binding affinities between Spy variants and Im7_{AAW} measured by intrinsic tryptophan fluorescence titration. (a-b) Intrinsic tryptophan fluorescence titration curves of Im7_{AAW} upon titration with different Spy variants that truncated at position 29 (a) or 27 (b). The intrinsic tryptophan fluorescence intensity at 349 nm was recorded with an excitation wavelength at 295 nm at 25 °C. (c) Binding affinities of Spy variants towards Im7_{AAW}. One-site bimolecular binding equation was used to fit the data to obtain the K_d values between Spy variants and Im7_{AAW}. Error bars indicate \pm s. d. of fitting.

2) The concentration of Spy used in the experiment shown in Figure 2g should be specified in the figure and the caption.

Our response: We have now specified Spy concentrations both in Fig. 2f and 2g.

Figure 2

3) The interpretation of Figures 5 and 6 would be greatly facilitated with a scheme illustrating the relative position of helices 1 to 4 in the structure of Spy.

Our response: We have now added the cartoon structure of Spy both in Fig. 5 and Fig. 6, with helices 1 to 4 labeled.

Figure 5

4) Two observations suggest that although electrostatic interactions play an important role there are other interactions between the N-terminal region and the cleft: mutation of residues 25 and 26 to R and K does not lead to an NMR spectrum indicating lack of interaction (Fig. 5c,f) and titration of the protein devoid of the N terminus with N-terminal peptide with position Asp 26 mutated to R or K slightly weakens the interaction (Fig. S7) but does not abolish it. The authors should discuss these results explicitly and provide possible explanations for them.

Our response: We agree with the reviewer that there are other interactions between the N terminus and the Spy cavity. We also did not mean to imply that the electrostatic interactions between residue D26 and Spy's concave surface were the only forces promoting client release.

In addition to the evidence pointed out by the reviewer, in Fig. 3 we showed that residues D2 and D10 can also approach the concave surface of Spy although not as effectively as residue D26. These interactions mediated by residues D2 and D10 may also contribute to the contact between the N terminus and Spy's concave surface. This could be a possible explanation for the residual interaction between the Nt-pep D26K/D26R and Spy₂₉₋₁₂₄ as observed in Supplementary Fig. 9.

In addition, we also considered whether the four hydrophobic residues M14, M15, M27, and M28 on the N terminus could form hydrophobic interactions with the two hydrophobic patches on the concave surface of Spy (Quan et al, *elife*, 3:e01584, 2014; He et al, *J Biol Chem*, 295(42):14488-14500, 2020). In Supplementary Fig. 5 we found that the M14 and M15 were relatively distant from Spy's concave surface, while in Fig. 1 we found that the absence of M27 and M28 did not affect the affinity and activity of Spy for client proteins. Therefore, we speculate that these hydrophobic residues may not be strongly involved in the interactions between the N terminus and the Spy cavity.

However, when the N terminus loses conformational constraints, as in the case of Fig. 6 and Supplementary Fig. 9, we cannot exclude the possibility that these hydrophobic residues also interact with the Spy cavity.

We thank the reviewer for the question, which will provoke further investigations to elucidate different contributions of charged and hydrophobic terminal residues in the intramolecular interaction and client release mechanism of Spy. As a clear explanation of the origin of the additional interactions would require considerable work, we prefer to keep relevant experiments and discussions as a topic for future studies.

Reviewer #2 (Remarks to the Author):

In this manuscript the teams of Shu Quan and Yi Xue investigate how intrinsically disordered regions of chaperone proteins affect their function, building on kinetic and structural information. The work focuses on the periplasmic protein chaperone Spy and the step of substrate release. Given that Spy is ATP-independent, the mechanistic impact of this work is very high, as in the absence of a coupled ATP hydrolysis cycle, regulation of the progression through different steps of cycle is not apparent. Therefore, detailed dynamic/thermodynamic/structural studies are required to understand the steps of substrate entry and release from the chaperone.

The authors utilize a broad array of techniques to obtain mechanistic understanding of substrate binding and release from Spy including: (i) a functional -folding- assay, (ii) NMR spectroscopy (with PREs), which is the most appropriate approach to capture the dynamic interaction of the N terminus with the concave surface of Spy, (iii) a very large number of mutants (deletion constructs, plus multiple point mutants), (iii) reciprocal mutation of charged residues in the N terminus and the concave, substrate binding, surface of Spy (that's the most elegant part of the work that demonstrates very nicely the proposed mechanism) and (iv) data with the N-terminal peptide.

The manuscript is very well written and all the conclusions are nicely supported.

Overall, this is a very significant piece of work in the field of chaperone biology and protein folding, which deserves publication at Nature Communications. Minor revisions are suggested below.

Our remarks to reviewer 2: thank you for your kind comments! Below, I respond to all your comments on a point-by-point basis.

Minor comments:

1) Line 201: substitute has for have

Our response: We have corrected this error.

2) The data presented in Fig. 5b, e, h make a better fit with the data presented in Fig. 3. Instead, it seems that the authors have segregated data based on the approach utilized (PREs vs. CSP). Still the flow of the text does makes sense as presented now. Looking at the PREs (Fig 3) my first thought was “what if the sticky MTSL binds non-specifically to the client binding site?” and “why simple spectrum overlays of the different constructs is not presented in addition to PREs to clarify this?”. Then, in Fig. 5 I see the data...

Our response: We are happy that reviewer 2 find the text does make sense as presented now.

3) The authors may want to present data (HSQC overlays or functional assays or kinetic measurements) acquired at different (higher) ionic strength. This excludes the possibility of affecting the structure of Spy upon any of the modifications utilized in here.

Our response: We have now performed additional NMR experiments to obtain the chemical shift perturbations (CSPs) of amide moieties of Spy₁₋₁₂₄ and Spy₂₉₋₁₂₄ at different ionic strengths (50 mM, 150 mM, and 250 mM NaCl). We found that CSP patterns caused by the existence of N terminus remain largely unchanged at various ionic strengths. This result indicated that the pattern we initially obtained at the physiological salt concentration, which reflects the contact patterns of the N terminus on the Spy cavity, is conserved over a wide range of salt concentrations (see figure below). We also noted that CSPs of some residues decreased with increasing ionic strength, implying a decreased tendency for the N terminus to reach the Spy cavity at higher ionic strengths. These residues are primarily located in the region of residue 50-65, which are, interestingly, featured by those positively charged key residues (K54, R55, R61, and R62). Since salt screening effect is known to weaken electrostatic interactions (Perez-Jimenez et al., *Biophys J*, 86(4):2414-29, 2004; Schreiber et al, *Chem Rev*, 109(3):839-60, 2009; Koldewey et al., *Cell*, 166(2):369-379, 2016; Lee et al., *Microbiology*, 164(7):992-997, 2018), this result strengthens our conclusion that the N terminus interacts with Spy cavity mainly through electrostatic interactions.

To more directly assess the effect of mutations introduced during PRE experiments or Spy characterization on the structure of Spy, we superimposed the HSQC spectra of Spy₁₋₁₂₄, Spy₁₋₁₂₄_{T5C}, and Spy₁₋₁₂₄_{CI} and found that they are very similar except for the peaks of the mutation sites and their nearby residues (see figure below). This suggests that the mutation on the N terminus did not cause substantial structural changes to the Spy cavity. Regarding mutagenesis in the Spy cavity, the resemblance of HSQC spectra between WT and the mutants indicates that the main body of Spy did not undergo remarkable conformational change upon these mutations (otherwise, the transfer of resonance assignment was impossible). In addition, the circular dichroism (CD) spectra of Spy₂₉₋₁₂₄_{K54D R55D}, Spy₂₉₋₁₂₄_{R61D R62D}, and Spy₂₉₋₁₂₄_{R89D} almost overlap with that of Spy₂₉₋₁₂₄, indicating that the mutations of K54, R55, R61, R62, and R89 on the Spy cavity also did not strongly change the

structure of Spy. Thus, we can exclude the possibility of significantly affecting Spy structure upon the modifications utilized in this work.

Figure legend: Mutation of the N terminus and the structural region of Spy does not have a strong effect on the structure of Spy. (a) CSPs of amide moieties of Spy₂₉₋₁₂₄ upon fusion of the N terminus in the presence of 50 mM (gray), 150 mM (orange), and 250 mM NaCl (blue). (b) An overlay of the 2D [¹⁵N, ¹H]-TROSY spectra of 400 μM [¹⁵N]-labeled Spy₁₋₁₂₄ (cyan) and 400 μM [¹⁵N]-labeled Spy_{1-124 T5C} (magenta). (c) An overlay of the 2D [¹⁵N, ¹H]-TROSY spectra of 400 μM [¹⁵N]-labeled Spy₁₋₁₂₄ (cyan) and 400 μM [¹⁵N]-labeled Spy_{1-124 Cl} (orange). (d) CD spectra of Spy₂₉₋₁₂₄ and Spy₂₉₋₁₂₄ mutants.

4) The last paragraph -as is- does not add anything to the manuscript. On the contrary, it makes it weaker. Of course, tethering the N terminus to the main body of Spy has a tremendous functional impact just because of the “tethering” effect, what the authors describe as “increases the effective local concentration”. Obviously, this imposes particular conformational constraints that affect the interaction compared to observations made with the peptide added in trans.

Looking at Fig. 5e, the CSP for H1, the H1-H2 loop and H4 are comparable to that shown in Fig. 6a, while the large CSP for residues before 35 can still attributed to the tethering effect. The large CSP shown in Fig. 6a for other regions may just be the result of 40x excess, where non-specific binding may appear having a similar effect (or even more pronounced) as a weak, but specific binding.

Our response: We agree with reviewer 2 and cannot exclude the possibility that non-specific binding may be involved in the weak binding between the Spy cavity and the free Nt-pep. We have now added this alternative interpretation as a note in the corresponding Supplementary figure legend as below.

“Note: Titration of Nt-pep to saturate Spy₂₉₋₁₂₄ was not feasible due to the solubility limitation of Nt-pep. Therefore, the K_d value can only be considered as an estimate and the influence of non-specific binding between Nt-pep and Spy₂₉₋₁₂₄ cannot be completely excluded.”

In addition to illustrating the importance of conformational constraints on the role of the N terminus, we performed these experiments in Fig. 6 to provide a possible explanation as to why Spy homologs chose to maintain the disordered N termini during evolution rather than developing a co-chaperone with Nt-pep-like properties to facilitate client release. We believe that this evolutionary perspective can enhance the impact of our findings and provoke reflection on many fundamental questions in chaperone biology. For all these reasons, we would like to keep this part. We have now added additional sentences in the discussion section on page 20:

“Interestingly, the peptide derived from the N terminus of Spy (Nt-pep) has much less ability to facilitate client release (Fig. 2f, g, Fig. 6, and Supplementary Fig. 9). While this may help explain why Spy homologs have chosen to maintain disordered N termini during evolution rather than develop co-chaperones with N terminus-like properties, it also raises more fundamental questions in chaperone biology that need to be addressed. For example, what are the minimum requirements for a prototypical chaperone that performs holding and folding activities in an ATP-independent manner? How can this knowledge guide the *de novo* design of artificial chaperones for a variety of cellular tasks? We believe that a deeper characterization of other simple chaperone systems will help to answer these questions.”

Reviewer #3 (Remarks to the Author):

In the manuscript entitled “An Active client protein release mechanism of an ATP-independent chaperone”, He et al. show that the N-terminal disordered sequence of the periplasmic, ATP-independent chaperone Spy facilitates client release. By using a combination of kinetic measurements, NMR spectroscopy, and molecular dynamics simulations, the authors studied the interactions between Spy (and several mutants) with their substrates and observed that the disordered N-terminal sequence of Spy facilitates substrate release from its positively charged, concave substrate-binding region. The authors propose that this release operates mainly via a “dynamic competition” mechanism mediated by electrostatic interactions, and speculate that this type of “release” can be a general mechanism used by ATP-independent chaperones to actively release their bound substrates.

I enjoyed reading this manuscript, and I think that the reported findings are novel. The experiments are clever and rigorously performed. However, the way some results are interpreted is confusing. I have a main “question” related to this that would need to be addressed in order to clarify the interpretation of the results (see below).

Our remarks to reviewer 3: Thank you for your careful review of our manuscript, and we are pleased that you find our findings novel and our experiments clever and rigorous. Below, I respond to all your comments on a point-by-point basis.

Major point

The authors show that the N-terminal sequence of Spy can, because its electrostatic interactions with the “positively charged” concave cavity that binds substrates, compete off the bound substrates. This means that the effect of the N terminus on substrate release operates via a “conformational selection” by binding to the available surface when the substrate is NOT bound to the cavity. If this was the case, then the effect of the N terminus on substrate binding should be observed in BOTH, the k_{on} and the k_{off} for substrate binding (not ONLY k_{off} as observed).

Moreover, the authors propose, based on the molecular simulations, that the N-terminal sequence is “attracted” by the positively charged surface that binds the substrates. The authors perform these simulations using the “empty” Spy. Again, if this effect can be seen in the substrate-free Spy, the competition effect should be observed both in k_{on} and the k_{off} .

Because the “free” N terminus sequence added in trans as a peptide does NOT have the same effect as the N terminus in the full length Spy, the authors propose that “the tethering of the N terminus to the structured region of Spy not only increases the effective local concentration of the N terminus but also imposes conformational constraints on its movement, making it easier to reach the central cavity of the concave surface and thus, more effective to compete with the client”. I was very pleased to read this (although at the very end of the results, this result should be moved up in the results section!) but I am not convinced by the interpretation: If the sequence of the N-terminal sequence in the context of the full length Spy binds the substrate-binding cavity and compete off the substrate, again, this effect should be also observed in the k_{on} of the substrate (in the presence AND the absence of the N terminus in Spy).

An alternative interpretation could be that, when substrate is BOUND, the chaperone undergoes a conformational change that makes the N terminus sequence more effective in competing off the substrate (restricts motion orienting it optimally?). In this case, the N terminus would not very have an effect on k_{on} . But after substrate binding, the N terminus could be differently positioned, being now able to effectively compete off the bound sequence.

Can you show this by NMR? Is substrate-bound Spy amenable for NMR studies?

Our response: We thank reviewer 3 for giving such an insightful alternative interpretation. We agree that the N terminus may exhibit some “conformational selection” behavior that allows it to bind better to the cavity of apo Spy than to the substrate-bound Spy. While it seems counterintuitive that this behavior does not alter k_{on} , a plausible explanation could be that the reduction in k_{on} for the substrate due to the intramolecular interaction may be offset by electrostatic attractions between the

N terminus and substrates.

The k_{on} rate between Spy and Im7 is diffusion controlled and is mainly influenced by long-range electrostatic interactions (Schreiber et al, Chem Rev, 109(3):839-60, 2009; Koldewey et al., Cell, 166(2):369-379, 2016; He et al, J Biol Chem, 295(42):14488-14500, 2020). When the N terminus binds to Spy cavity, on the one hand, some of the positively charged residues in the cavity are neutralized by the N-terminal residues D26, D2, and D10, and the attraction to the substrate is weakened. On the other hand, the N-terminal residues K12, H16, H17, K18, and K20 can also provide additional positive charges to attract negatively charged substrates. We speculate that these contradictory effects of the N terminus on the electrostatic potential of the Spy cavity could largely cancel each other. Therefore, when the N terminus is bound to Spy cavity, the electrostatic potential of the Spy cavity is similar to that of the cavity when the N terminus is not bound, resulting in similar k_{on} rates.

To test this hypothesis, we decreased the positive charge of the N terminus by mutating residues H16, H17, and K18 to alanine and measured the binding kinetics of the resulting mutants towards the client Im7_{AAW}. We found that the k_{on} rate of Spy decreased from $2.76 \times 10^5 \text{ M}^{-1}\text{s}^{-1}$ to $1.83 \times 10^5 \text{ M}^{-1}\text{s}^{-1}$ due to alanine substitutions. Further reduction of the positive charges on the N terminus by mutating residues H16, H17, and K18 to aspartate resulted in the decrease of k_{on} rate by a factor of 8. In addition, the k_{off} rates of mutants Spy_{H16A H17A K18A} and Spy_{H16D H17D K18D} also increased by 3.5 and 6.4-fold, resulting in 5.3 and 51.0-fold increases in the K_{d} . Thus, these results together with the biophysical and biochemical characterizations in our manuscript emphasized the importance of charge and the interplay between the N terminus and the Spy cavity in both client association and dissociation.

Figure legend: Kinetic characterization of Spy^{H16A H17A K18A} and Spy^{H16D H17D K18D} towards the client Im7_{AAW} by BLI. (a-b) Association and dissociation kinetic curves of various concentrations of Spy^{H16A H17A K18A} (a) and Spy^{H16D H17D K18D} (b) towards Im7_{AAW}. Black curves indicated the fitting curves. (c) The association rate constants k_{on} , dissociation rate constants k_{off} , and dissociation constants K_d of Spy^{H16A H17A K18A} and Spy^{H16D H17D K18D}.

In addition, we find that the alternative interpretation proposed by reviewer 3 is likely to be plausible as well, as it has been previously reported that Spy slightly changes conformation upon substrate binding. Although comparison of the secondary backbone ¹³C chemical shifts of Spy₂₉₋₁₂₄ and Spy₂₉₋₁₂₄-Im7_{A3} suggests an overall similar secondary structure of Spy upon substrate binding (Supplementary Fig. 8c), Horowitz et al. showed that the substrate-bound Spy dimer twists 9° around its center compared with its client-free state. Moreover, the linker region consisting of residues 47-57 becomes mostly disordered, which is thought to contribute to Spy’s promiscuous recognition of various substrates (Scott et al., Nat Struct Mol Biol, 23(7):691-7, 2016).

To probe the conformational changes of Spy and its effect on self-binding of the N terminus by NMR, we performed intramolecular PRE experiments on ¹⁵N-labeled Spy₁₋₁₂₄ CI in the presence of the unlabeled client Im7_{A3}. We found that when Im7_{A3} was bound to Spy, the N terminus still

made transient contacts with the Spy cavity (see figure below), although the PRE intensity ratios were generally much smaller when Im7_{A3} was bound compared to the signals obtained on apo Spy.

Figure legend: Intramolecular PRE effects of the spin label attached to CI on 400 μM [^2H , ^{15}N]-labeled Spy₁₋₁₂₄ in the presence (blue) and absence (pink) of 400 μM unlabeled Im7_{A3}. Asterisks indicate residues without available PRE ratios due to peak overlapping and line broadening upon the binding of Im7_{A3}.

Unfortunately, due to peak overlapping and line broadening, the backbone assignments for some residues of Spy₁₋₁₂₄ in the presence of Im7_{A3} were not available. Thus, we were unable to assess whether the N terminus could be positioned differently upon substrate binding to compete more efficiently with substrate for Spy binding. As suggested by the reviewer, conformational changes of Spy upon substrate binding may result in the N terminus being more readily bound to certain residues on the Spy cavity. However, because of current technical limitations and the missing link between Spy conformational changes and N terminal positions, we are currently unable to fully test the reviewer 3's hypothesis. We can only confidently conclude that the N terminus is able to approach the cavity of Spy in the presence and absence of the substrate. Therefore, we prefer to make related experiments and discussions the subject of future investigation. We envision that ongoing high-resolution structural determinations, IDR simulations, and methods to measure kinetics of an intrinsic part of a protein will further elucidate the interpretation of Spy dynamics in the future.

Minor points

1. Can you describe a bit more the “super Spy” variant created previously?

Our response: We have now added the following sentence on pages 5 and 6 to describe the "super spy" variant.

“We previously isolated activity-enhancing “Super Spy” variants through a genetic selection that coupled Spy’s ability to stabilize an unstable mutant of Im7 with antibiotic resistance of *E. coli*²⁰. These “Super Spy” variants all sacrificed their own stability for flexibility, thereby enhancing

chaperone activity²⁰.”

2. Figures: I very much appreciate that the authors show raw data for the performed experiments. In some cases (for sure for Figure 1) the raw results could be moved to the supplementary material for clarity.

Our response: We have moved Fig. 1g, h, I, j to Supplementary Fig. 1 and updated the corresponding figure legends.

3. Please make the labels of the Y axis a bit larger.

Our response: We have now enlarged the labels of the Y axis of all figures.

4. Figure 3: not need to specify “180°” for each graph if they are all the same. Same with “residue number” labels (text and numbers). That would help with clutter (same in Fig 4).

Our response: We have now removed the redundant text and labels in Figures 3 and 4.

REVIEWERS' COMMENTS

Reviewer #1 (Remarks to the Author):

The authors have addressed most the concerns states in the first review. There are a few outstanding issues to address, however, but they are not major:

1 - I continue to have issues with the title of the paper and a statement in the abstract.

I do not consider it appropriate to have a title/abstract containing the words "Self-aided" referring to a protein. After having mentioned this in my first review I now consider this an editorial matter, however.

2 - In my second major issue I stated that the result that removal of residues 1 to 24 decreases the activity of Spy is not in agreement with the model put forward by the authors.

In their reply the authors insist that the remaining N-terminus (residues 24-28) is even more capable to compete against substrates than the complete N-terminus: an analysis of the sequence shows that residues 1 to 23 include a number of positively charged residues that may counteract the effect of Asp 26. I would suggest that the authors include such a comment in their second revision.

Reviewer #2 (Remarks to the Author):

Although I believe that segregating data based on the approach is not good the manuscript, it's only of cosmetic nature. The authors have fully addressed all of my comments. They have also incorporated a large set of new data and also present a more comprehensive analysis of the data included in the original manuscript. The paper is ready to be published to Nature Communications.

Reviewer #3 (Remarks to the Author):

Although in the response to reviewers the authors have answered my question, I think that if there are alternative explanations/models for the presented data they should be, at least briefly, discussed in the manuscript. Otherwise, I think it is suitable for publication.

REVIEWER COMMENTS

Reviewer #1 (Remarks to the Author):

The authors have addressed most the concerns states in the first review. There are a few outstanding issues to address, however, but they are not major.

Our response: thank you for your kind comments! Below, I respond to all your comments on a point-by-point basis.

Minor issues

1) I continue to have issues with the title of the paper and a statement in the abstract.

I do not consider it appropriate to have a title/abstract containing the words "Self-aided" referring to a protein. After having mentioned this in my first review I now consider this an editorial matter, however.

Our response: We have removed the "self-aided" language in the abstract and discussion section. In addition, we have changed the title as "Insights into the client protein release mechanism of the ATP-independent chaperone Spy".

2) In my second major issue I stated that the result that removal of residues 1 to 24 decreases the activity of Spy is not in agreement with the model put forward by the authors.

In their reply the authors insist that the remaining N-terminus (residues 24-28) is even more capable to compete against substrates than the complete N-terminus: an analysis of the sequence shows that residues 1 to 23 include a number of positively charged residues that may counteract the effect of Asp 26. I would suggest that the authors include such a comment in their second revision.

Our response: thank you for raising this valuable explanation! We have added this comment on

page 7 as below:

“The release promoting effect of residues H24, Q25, and D26 was even more pronounced after the removal of N-terminal residues 1-23 (**Fig. 1e**), suggesting that residues 1-23 are also able to counteract the effects of H24, Q25, and D26. Supporting this idea, we found five positively charged residues (K12, H16, H17, K18, and K20) located in the N terminal region prior to residues 24-26.”

Reviewer #2 (Remarks to the Author):

Although I believe that segregating data based on the approach is not good the manuscript, it's only of cosmetic nature. The authors have fully addressed all of my comments. They have also incorporated a large set of new data and also present a more comprehensive analysis of the data included in the original manuscript. The paper is ready to be published to Nature Communications.

Our response: we are grateful for your kind comments!

Reviewer #3 (Remarks to the Author):

Although in the response to reviewers the authors have answered my question, I think that if there are alternative explanations/models for the presented data they should be, at least briefly, discussed in the manuscript. Otherwise, I think it is suitable for publication.

Our response: thank you for your kind comments! We have now added these alternative explanations on page 17-18 as below:

“Interestingly, these intramolecular interactions do not obviously affect the client association rate (k_{on}). A plausible explanation for this seemingly counterintuitive observation is that upon binding to the Spy cavity, the N terminus not only neutralizes some of the positive charges on the cavity through residues D2, D10, and D26, but also provides additional positively charged residues (K12, H16, H17, K18, and K20) to attract negatively charged clients in the distance. These contradictory

effects may cancel each other out, rendering the net effect of the N terminus on the k_{on} rate negligible. Alternatively, since Spy has been previously reported to change conformation slightly upon client binding¹⁹, the N terminus may be positioned differently upon client binding in order to compete more efficiently with the client for Spy binding. If this is the case, the k_{on} rate is not necessarily affected by the N terminus before client binding. We envision that establishing high-resolution IDR structure determination and simulation methods will help further elucidate the mechanisms involved in the future.”